# Step-Resolved Data Attribution for Looped Transformers

**Georgios Kaissis** [* 1]  **David Mildenberger** [* 2 3]  **Juan Felipe Gomez** [* 4]  **Martin J. Menten** [2 3 5]  **Eleni Triantafillou** [6]

## Abstract

We study how individual training examples shape the internal computation of looped transformers, where a shared block is applied for $\tau$ recurrent iterations to enable latent reasoning. Existing training-data influence estimators such as TracIn yield a single scalar score that aggregates over all loop iterations, obscuring when during the recurrent computation a training example matters. We introduce *Step-Decomposed Influence (SDI)*, which decomposes TracIn into a length-$\tau$ influence trajectory by unrolling the recurrent computation graph and attributing influence to specific loop iterations. To make SDI practical at transformer scale, we propose a TensorSketch implementation that never materialises per-example gradients. Experiments on looped GPT-style models and algorithmic reasoning tasks show that SDI scales excellently, matches full-gradient baselines with low error and supports a broad range of data attribution and interpretability tasks with per-step insights into the latent reasoning process.

## 1. Introduction

The capability of Large Language Models (LLMs) to perform complex reasoning has become a central focus of current machine learning research. This shift is driven by the observation that allocating additional compute at test-time—often realised through Chain-of-Thought (CoT) prompting—can substantially improve performance on benchmarks requiring multi-step logic, mathematics, and algorithmic execution (Geiping et al., 2025; McLeish et al., 2025). However, standard CoT relies on explicit, discrete token generation, which introduces significant computational overhead and expressive limitations. Generating intermediate tokens in natural language is costly and often includes many tokens that contribute little to advancing the underlying reasoning state (e.g., formatting tokens or connective words), consuming both compute and context window budget (Chen et al., 2025). More fundamentally, requiring intermediate computation to be communicated through a discrete natural-language channel can be a low-bandwidth constraint relative to the model's continuous hidden representations, which may be a more suitable substrate for high-dimensional intermediate computation (Hao et al., 2024).

To overcome these limitations, recent work has pivoted toward *latent reasoning*, where the model performs intermediate computations in a high-bandwidth continuous space rather than a discrete token space (Zhu et al., 2025a; Saunshi et al., 2025). Among the architectures facilitating this paradigm, the *looped* (also called *weight-tied* or *depth-recurrent*) transformer has emerged as a particularly promising candidate (Yang et al., 2023a; Giannou et al., 2023). A looped transformer operates by applying a shared block of parameters recursively over its own activations (and possibly input embeddings) for a fixed or dynamic number of iterations; e.g., at a loop horizon $\tau = 8$, the same transformer block is applied eight times to the residual stream before producing final logits. This architectural design decouples the parameter count from the computational depth axis: a looped transformer can maintain a small memory footprint while allocating dynamic compute budgets to harder instances by simply looping more times (Fan et al., 2024; Bae et al., 2025). The efficacy of this approach has been demonstrated across domains, from algorithmic tasks (Yang et al., 2023b) to large-scale architectures like Ouro and RecurrentGemma (Zhu et al., 2025b; Botev et al., 2024), as well as specialised reasoning architectures like Tiny Recursive Models (Jolicoeur-Martineau, 2025) and Universal Reasoning Models (Gao et al., 2025).

**Why step-resolved data attribution?** Despite their growing prominence, the internal dynamics of looped transformers remain underexplored. While it is known that looping improves performance, understanding *how* latent states ("thoughts") evolve across loop iterations and how specific training data influences this evolution is an open challenge. Traditional data attribution methods, such as Influence Func-

---
[*]Equal contribution  [1]Hasso Plattner Institute for Digital Engineering, University of Potsdam, Potsdam, Germany [2]Technical University of Munich, Munich, Germany [3]Munich Center for Machine Learning (MCML), Munich, Germany [4]Harvard University, Cambridge, MA, USA [5]Imperial College London, London, United Kingdom [6]Google DeepMind, London, United Kingdom. Correspondence to: Georgios Kaissis <georg.kaissis@hpi.de>.

*Proceedings of the 43rd International Conference on Machine Learning*, Seoul, South Korea. PMLR 306, 2026. Copyright 2026 by the author(s).

tions (Koh & Liang, 2017) or TracIn (Pruthi et al., 2020), treat the model as a static function mapping input to output. While these estimators can be applied to looped architectures, their output is a *single scalar* influence score that aggregates over all applications of a shared block, and therefore does not localise influence to specific loop iterations. To thus rigorously interpret looped models, we require a granular view that apportions credit to the specific computational steps they perform. In looped transformers, the loop horizon $\tau$ is a *test-time compute knob*: increasing $\tau$ changes the amount of internal computation without changing the parameter set or the number of generated tokens. In this regime, practitioners may be interested not only in *which* training examples matter, but *when* they matter during the recurrent computation. A single scalar influence score cannot reveal whether an example primarily supports early iterations (e.g., parsing/grounding) versus late iterations (e.g., iterative refinement), nor whether positive and negative effects cancel across steps. Step-resolved attribution on the other hand can enable e.g., (i) **calibrating test-time compute** by identifying the "influence horizon" where training data ceases to impact latent state evolution, removing the need for expensive hyperparameter retraining; (ii) **detecting signal cancellation**, where a near-zero aggregate scalar score masks significant but opposing effects at early versus late iterations; and (iii) **depth-targeted data curation**, allowing practitioners to filter for examples that specifically drive iterative refinement rather than early input processing.

In this work, we introduce **Step-Decomposed Influence (SDI)**, a novel framework for interpreting looped transformers via *unrolled* data attribution. By generalising the TracIn estimator (Pruthi et al., 2020) to the recurrent setting, we decompose the scalar influence of a training point into a trajectory of influences across loop iterations. We provide a reference implementation of the SDI framework at https://github.com/gkaissis/step-decomposed-influence-oss.

**Contributions** Our specific contributions are as follows:

- **Step-Decomposed Influence (SDI):** We formalise the attribution of training data to internal recurrent steps of weight-tied models. We prove a conservation identity showing that SDI losslessly decomposes standard TracIn, bridging the gap between static influence scores and dynamic latent computation.

- **Streaming TensorSketch with Tighter Bounds:** We address the memory bottleneck of gradient-based attribution via a *streaming sketch-during-backprop* algorithm that avoids materialising per-example gradients. Theoretically, we derive tight variance bounds for TensorSketch on outer-product sums that are strictly tighter than prior art, ensuring unbiased, low-variance estimation at transformer scale.

- **Applications to Latent Reasoning:** We demonstrate SDI's utility across three distinct regimes: recovering finite-state automata circuits in parity tasks, linking test-time compute scaling to late-stage influence in Sudoku, and revealing implicit geometric growth of influence in a looped 330M parameter LLM (Nanochat), suggesting that looped transformers spontaneously learn to represent their own progress through the recurrence.

Figure 1 gives a compact overview of the SDI pipeline.

**Conflict of Interest Disclosure** The authors have no financial conflicts of interest to disclose.

## 2. Background

In this section we give a brief overview of looped transformer models and how they are trained for language modelling tasks. We also overview the TracIn estimator (Pruthi et al., 2020) and its application for computing the influence of training examples on predictions of the model.

**Looped transformers** The parameters $\mathbf{w}$ of a looped transformer are partitioned into three functional blocks: a *read-in* block $\mathbf{w}_{\text{in}}$, a recurrent *body* $\mathbf{w}_{\text{body}}$, and a *read-out* block $\mathbf{w}_{\text{out}}$. Given an input sequence $\mathbf{x}$, the read-in block maps sequences to an embedding $\mathbf{h}_0 = E(\mathbf{x}; \mathbf{w}_{\text{in}})$. This embedding has shape $\mathbb{R}^{L \times d}$ for sequence length $L$ and hidden size $d$. The body block then iteratively refines the embedding for $\tau$ steps by looping over activations: $\mathbf{h}_t = F(\mathbf{h}_{t-1}, \mathbf{e}_{\text{inj}}; \mathbf{w}_{\text{body}})$ for $t \in \{1, \ldots, \tau\}$. We will often refer to $\tau$ as the depth dimension and $t$ as a "step". To stabilise training across steps, looped transformers often employ input injection, represented above as the dependence on $\mathbf{e}_{\text{inj}}$ (Bansal et al., 2022). This input injection can be additive as in Fan et al. (2024), concatenative as in Geiping et al. (2025) or null. The read-out block maps the final activation $\mathbf{h}_\tau$ to logits $y = R(\mathbf{h}_\tau; \mathbf{w}_{\text{out}})$, and standard cross-entropy loss is used to train the looped model. It is common for looped models to be trained with randomised depths (Geiping et al., 2025) (sampling $\tau$ from a probability distribution) or with input-dependent $\tau$ (Fan et al., 2024) to encourage robustness to changing $\tau$, as is often the case at inference-time to optimize test-time compute.

**TracIn** To quantify the effect of training data on model predictions, we build upon the TracIn estimator (Pruthi et al., 2020). TracIn approximates the influence of a train example $z$ on an example $z'$ by summing the dot product of their loss gradients over saved checkpoints $\mathcal{W} = \{\mathbf{w}_k\}_{k=1}^K$. Letting $\ell$ be the loss function (e.g., cross-entropy); the TracIn estimator is given by:

$$\text{TracIn}_{\mathbf{w}}(z, z') = \sum_{k=1}^K \eta_k \nabla_{\mathbf{w}} \ell(\mathbf{w}_k; z) \cdot \nabla_{\mathbf{w}} \ell(\mathbf{w}_k; z'). \quad (1)$$

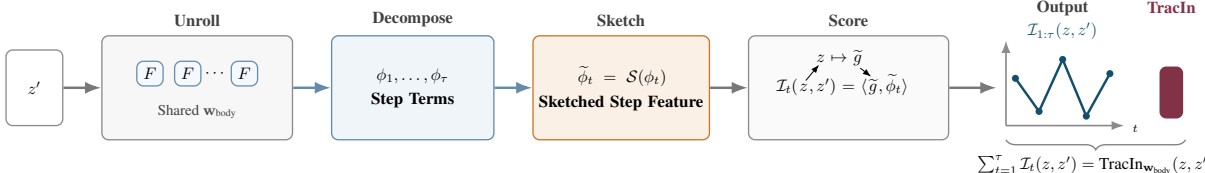

*Figure 1.* **SDI overview.** A query example $z'$ is run through the shared recurrent body, the body gradient is decomposed into step terms $\phi_t$, each term is sketched during backpropagation, and dot products with a sketched gradient $\tilde{g}$ of a train example $z$ yield the stepwise influence trajectory $\mathcal{I}_{1:\tau}(z, z')$. Summing the trajectory recovers scalar body-parameter TracIn.

Here $\eta_k$ denotes the learning rate associated with checkpoint $\mathbf{w}_k$. Note that while the gradient with respect to matrix parameters is a matrix, TracIn flattens all gradients into vectors to compute the dot product. Intuitively, for fixed $z'$, training examples $z$ with $\text{TracIn}(z, z')>0$ reduce the loss on $z'$ along the training trajectory, while $z$ with $\text{TracIn}(z, z')<0$ increase the loss on $z'$. Interpreting $\text{TracIn}(z, z')$ as the "influence" of train example $z$ on example $z'$, we follow Feldman (2020); Feldman & Zhang (2020) and distinguish between *self-influence* $(z = z')$, a proxy for memorisation during training, and *cross-influence* $(z{\neq}z')$, which captures how training on $z$ relates to generalisation on a held out example $z'$. Our work exploits the recurrent structure of $\mathbf{w}_{\text{body}}$ to decompose the TracIn gradients and therefore the estimator across steps, which we outline in the next section. For a discussion on why we chose TracIn as our influence proxy over other approaches such as Influence Functions, see Appendix A.

## 3. Step-Decomposed Influence

We wish to apply and build on the TracIn estimator introduced in Section 2 to looped transformers. A naïve application of Equation (1) would use the total gradient $\nabla_{\mathbf{w}_{\text{body}}}\ell$, which unrolls into the sum the contribution of the gradients across steps. Using this for TracIn obscures whether a training example $z$ influences $z'$ in early vs. late loop steps. To resolve this, we introduce **Step-Decomposed Influence (SDI)**, which turns a single scalar influence score into a *step-resolved influence trajectory*. This enables analyses that are not accessible from scalar influence estimators, such as localising which training examples affect *early* vs. *late* loop steps and identifying where additional steps yield diminishing returns.

**Gradients as a sum over steps:** Let $\ell(\mathbf{w}; z)$ denote the loss of a model with parameters $\mathbf{w}$ at an example $z$; we omit explicit dependence on $(\mathbf{w}; z)$ when unambiguous. Recall that the per-step activations are given by $\mathbf{h}_t = F(\mathbf{h}_{t-1}, \mathbf{e}_{\text{inj}}; \mathbf{w}_{\text{body}})$ for $t \in \{1, \ldots, \tau\}$. The dependence of $\mathbf{h}_t$ on $\mathbf{w}_{\text{body}}$ and on $\mathbf{h}_{t-1}$, which also depends on $\mathbf{w}_{\text{body}}$, turns the total derivative of the loss with respect to $\mathbf{w}_{\text{body}}$ into a sum of Vector-Jacobian Products.

**Proposition 1.** For a looped transformer with $\tau$ steps and sequence length $L$, let $\mathbf{h}_{t,j} \in \mathbb{R}^d$ denote the hidden state of the $j$-th token at step $t$. The *total derivative* of the loss with respect to $\mathbf{w}_{\text{body}}$ can be unrolled into a sum over $\tau$ steps and $L$ tokens:

$$\phi_t = \sum_{j=1}^{L} \frac{\mathrm{d}\ell}{\mathrm{d}\mathbf{h}_{t,j}} \frac{\partial \mathbf{h}_{t,j}}{\partial \mathbf{w}_{\text{body}}}, \qquad \frac{\mathrm{d}\ell}{\mathrm{d}\mathbf{w}_{\text{body}}} = \sum_{t=1}^{\tau} \phi_t \qquad (2)$$

Where $\frac{\mathrm{d}\ell}{\mathrm{d}\mathbf{h}_{t,j}} \in \mathbb{R}^d$ is the gradient at step $t$ and token $j$ computed via backpropagation through time (BPTT), and $\frac{\partial \mathbf{h}_{t,j}}{\partial \mathbf{w}_{\text{body}}} \in \mathbb{R}^{d \times |\mathbf{w}_{\text{body}}|}$ is the Jacobian of the function $F$ with respect to the parameters $\mathbf{w}_{\text{body}}$, while holding the input state $\mathbf{h}_{t-1}$ and input injection $\mathbf{e}_{\text{inj}}$ constant.

Moreover, if $\mathbf{w}_{\text{body}}$ is a concatenation of a matrix $\mathbf{W}$ and a vector $\mathbf{b}$, and in $F$ they appear only through a tokenwise linear map of the form $\mathbf{c}_{t,j} = \mathbf{W}\, \mathbf{a}_{t,j} + \mathbf{b}$ where $\mathbf{a}_t = f(\mathbf{h}_{t-1})$ is the forward activation, then $\phi_t$ simplifies to sums of outer products for $\mathbf{W}$ and to sums of vectors for $\mathbf{b}$:

$$\delta_{t,j} = \frac{\mathrm{d}\ell}{\mathrm{d}\mathbf{c}_{t,j}} \qquad (3)$$

$$\phi_t^{\mathbf{W}} = \sum_{j=1}^{L} \delta_{t,j} \otimes \mathbf{a}_{t,j}, \qquad \phi_t^{\mathbf{b}} = \sum_{j=1}^{L} \delta_{t,j}. \qquad (4)$$

Let $\mathbf{W} \in \mathbb{R}^{d_{\text{out}} \times d_{\text{in}}}$, then $\delta_{t,j} \in \mathbb{R}^{d_{\text{out}}}, \mathbf{a}_{t,j} \in \mathbb{R}^{d_{\text{in}}}$. Moreover $\phi_t^{\mathbf{W}} \in \mathbb{R}^{d_{\text{out}} \times d_{\text{in}}}, \phi_t^{\mathbf{b}} \in \mathbb{R}^{d_{\text{out}}}$, and $\phi_t = (\phi_t^{\mathbf{W}}, \phi_t^{\mathbf{b}})$ with $|\mathbf{w}_{\text{body}}| = d_{\text{out}}d_{\text{in}} + d_{\text{out}}$.

If $F$ instead contains many modules each with matrix-valued and vector-valued parameters of the same form (e.g. $F$ is a GPT-2 small model), then we apply the above analysis individually to each matrix and vector parameter, and concatenate to form $\phi_t$.

*Proof.* By definition of the total derivative. $\square$

**Step-Decomposed Influence (SDI)** Recall that TracIn estimates the influence of a training example $z$ on an example $z'$ by summing gradient dot products over checkpoints (cf. Equation (1)). SDI refines this by localising the contribution to a specific step $t$. Concretely, define the step-localised influence

$$\mathcal{I}_t(z, z') := \sum_{k=1}^{K} \eta_k \nabla_{\mathbf{w}_{\text{body}}} \ell(\mathbf{w}_k; z) \cdot \phi_t(\mathbf{w}_k; z'). \qquad (5)$$

As a direct consequence of Proposition 1, this step-localized cross influence losslessly decomposes TracIn across steps:

$$\text{TracIn}_{\mathbf{w}_{\text{body}}}(z, z') = \sum_{t=1}^{\tau} \mathcal{I}_t(z, z'), \qquad (6)$$

We are now ready to formally define SDI.

**Definition 1 (Step-Decomposed Influence (SDI)).** Given checkpoints $\mathcal{W} = \{\mathbf{w}_k\}_{k=1}^K$ and learning rates $\{\eta_k\}_{k=1}^K$, the SDI trajectory of a training example $z$ on a test example $z'$ through the step trajectory is the length-$\tau$ vector

$$\text{SDI}(z, z') := (\mathcal{I}_t(z, z'))_{t=1}^{\tau}. \qquad (7)$$

SDI is generally asymmetric in $z$ and $z'$. Our default choice decomposes the *test-side* recurrence and answers: *How much did training on $z$ (across all its training-time loop steps) influence loop step $t$ of the computation on $z'$?* Other decompositions are possible, depending on whether one wants to attribute *effects on test-time steps* (as above) or *causes from training-time steps*. For example, a training-step decomposition can be defined as

$$\widehat{\mathcal{I}}_t(z, z') := \sum_{k=1}^K \eta_k \, \phi_t(z; \mathbf{w}_k) \cdot \nabla_{\mathbf{w}_{\text{body}}} \ell(\mathbf{w}_k; z'). \quad (8)$$

More granularly, one can form a stepwise influence matrix

$$\mathcal{I}_{s,t}(z, z') := \sum_{k=1}^K \eta_k \, \phi_s(z; \mathbf{w}_k) \cdot \phi_t(z'; \mathbf{w}_k). \qquad (9)$$

This matrix view makes the "train-time step" vs. "test-time step" distinction explicit. Although these decompositions are linear in the step index, each $\phi_t$ contains the BPTT cotangent $d\ell/d\mathbf{h}_{t,j}$, which depends on all downstream loop applications through the unrolled computation graph, together with the local Jacobian at the $t$-th reuse of the shared block. Thus, an early structural effect that changes later latent states is still reflected in the later step terms that receive downstream credit; SDI only chooses where in the shared-parameter recurrence to book that credit.

**Sketch-during-backprop** Computing and storing the full-dimensional vectors $\phi_t(\mathbf{w}_k; z') \in \mathbb{R}^{|\mathbf{w}_{\text{body}}|}$ for every step, checkpoint and example is prohibitively expensive. Prior TracIn implementations typically follow a *project-after-the-fact* pipeline: first compute and store full per-example gradients, then apply a random projection to reduce dimensionality (Pruthi et al., 2020; Hu et al., 2025). As per-example gradients are never materialized in standard training pipelines (the sums in Proposition 1 and batch reduction of the loss are done in parallel in backwards-mode automatic differentiation), this approach retains the peak memory and runtime burden of per-example gradient materialisation. In contrast,

SDI uses a *sketch-during-backprop* pipeline: we compute projected/sketched features *on the fly* and *never instantiate the full per-example gradients*. This efficient implementation is one of our core contributions that unlocks computing SDI and TracIn at transformer scale and can also be used more broadly whenever *on-the-fly compressed per-example gradients* are required.

We rely on two sketching primitives: for a vector $\mathbf{x} \in \mathbb{R}^d$, CountSketch (Charikar et al., 2002) defines a linear map $\text{CS} : \mathbb{R}^d \to \mathbb{R}^m$ using a hash $h : [d] \to [m]$ and signs $s : [d] \to \{\pm 1\}$: $(\text{CS}(x))_j := \sum_{i : h(i) = j} s(i) \, \mathbf{x}_i$. This is a sparse random projection that preserves inner products in expectation while requiring only $\mathcal{O}(d)$ computation to apply. CS applies to vectors, and can therefore be used to sketch gradients of vector parameters such as biases $\mathbf{b}$ in Proposition 1 or affine parameters in normalisation layers. The dominant parameters in transformer blocks, however, are matrix weights (e.g., the linear maps in attention and MLPs), whose per-example gradients factor as sums of outer products of backpropagated signals $\delta_{t,j}$ and activations $\mathbf{a}_{t,j}$ as shown in Proposition 1.

TensorSketch builds off of CS and provides an efficient sketch for such outer products by convolving the CountSketch of the two vectors (Pham & Pagh, 2025). Formally, for $\mathbf{u} \in \mathbb{R}^{d_{\text{out}}}$ and $\mathbf{v} \in \mathbb{R}^{d_{\text{in}}}$, TensorSketch defines a map $\text{TS} : \mathbb{R}^{d_{\text{out}}} \times \mathbb{R}^{d_{\text{in}}} \to \mathbb{R}^m$ such that $\text{TS}(\mathbf{u}, \mathbf{v})$ is a CountSketch of the tensor product $\mathbf{u} \otimes \mathbf{v}$ computed in $\mathcal{O}(d_{\text{out}} + d_{\text{in}} + m \log m)$ time. Since TensorSketch is linear, it can be applied to sums of outer products like gradients from matrix parameters. See Appendix B for a more detailed explanation of CountSketch and TensorSketch. In our implementation, we apply CS/TS independently (meaning the underlying hashes are sampled independently) at each parameter tensor then concatenate to obtain a sketch of the full per-example gradient. Formally, let $\mathcal{S}_m(\cdot)$ denote the block concatenation of independent CS/TS maps, each with output dimension $m$, applied per parameter tensor (vector parameters via CS, matrix parameters via TS), producing a single *global* sketched vector. Note that if some gradient $\mathbf{g}$ contains $\alpha$ total parameter gradients, then $S_m$ outputs an $\alpha m$-dimensional vector. Since CS and TS are linear, so is $\mathcal{S}$. We denote sketched vectors by $\widetilde{\mathbf{u}} := \mathcal{S}(\mathbf{u})$ and estimate SDI in the sketched space:

$$\widetilde{\mathcal{I}}_t(z, z') := \sum_{k=1}^K \eta_k \, \widetilde{\nabla_{\mathbf{w}_{\text{body}}} \ell(\mathbf{w}_k; z)} \cdot \widetilde{\phi_t(\mathbf{w}_k; z')}. \quad (10)$$

By linearity of $\mathcal{S}$ and Equation (2), we have that:

$$\widetilde{\nabla_{\mathbf{w}_{\text{body}}} \ell(\mathbf{w}_k; z)} = \sum_{t=1}^{\tau} \widetilde{\phi_t(\mathbf{w}_k; z')}. \qquad (11)$$

Since TS is a function that acts directly on the vectors that form the outer product, and not the outer product itself, our

approach computes these sketches *directly during backprop-agation* using intermediate signals ("hooks") similar to the approach in Yousefpour et al. (2021). Recall that the hidden step states are given by $\mathbf{h}_t = F(\mathbf{h}_{t-1}, \mathbf{e}_{\text{inj}}; \mathbf{w}_{\text{body}})$ for $t \in \{1, \ldots, \tau\}$. For each step $t$, token $L$, and matrix parameter in $F$, SDI applies TensorSketch to the cached forward activation $\mathbf{a}_{t,j}$ and the backpropagation signal $\delta_{t,j}$. It then sums the sketched outer products over tokens $j$ to form $\tilde{\phi}_t$ before concatenating sketches across parameter tensors to form a single global feature vector per example and step. Algorithm 1 summarises the approach.

From a systems perspective, the sketches require $\Theta(B\tau m)$ extra storage per parameter-tensor for storing $\tilde{\phi}_{1:B,t}$ for batch size $B$, horizon $\tau$, and sketch dimension $m$. Note that materializing the full per-example $\phi_{1:B,t}$ would scale as $\Theta(B\tau|\mathbf{w}_{\text{body}}|)$. In our 135.1M-parameter GPT-2 experiment, $|\mathbf{w}_{\text{body}}|$ is on the order of 100M, whereas $m = 2048$ with 48 parameter tensors, therefore SDI is $\sim 1,000\times$ less memory intensive. Computationally, our procedure introduces only an additive overhead to standard BPTT—dominated by the FFT-based TensorSketch updates and scaling linearly in the number of per-step module invocations (hence in $B\tau$) and as $m \log m$ in the sketch dimension—but it is incurred inline during the same backward pass. Empirically, this overhead is small at transformer scale as seen in the experimental section.

---

### Algorithm 1

**Require:** Checkpoint $\mathbf{w}$; batch $Z = \{z_i\}_{i=1}^B$; loop horizon $\tau$; sketch dim. $m$; per-parameter sketch maps $\{\mathsf{CS}, \mathsf{TS}\}$; sequence length $L$; recurrent function $F$
1: $\tilde{\phi}_{1:B,t} \leftarrow [\,]$ for all $t = 1, \ldots, \tau$
2: Attach hooks to $F$ to cache forward inputs $\mathbf{a}_{t,j}$ and read backward signals $\delta_{t,j}$
3: Run one forward pass
4: **for** each step $t$ during backward pass **do**
5:     **for** each matrix/vector parameter $W, b$ in $F$ **do**
6:         $\widetilde{\nabla_W}\ell = \sum_{j=1}^L \mathsf{TS}(\delta_{t,j}, a_{t,j})$
7:         $\widetilde{\nabla_b}\ell = \sum_{j=1}^L \mathsf{CS}(\delta_{t,j})$
8:         Append $(\widetilde{\nabla_W}\ell, \widetilde{\nabla_b}\ell)$ to $\tilde{\phi}_{1:B,t}$
9:     **end for**
10: **end for**
11: $\tilde{\mathbf{g}}_{1:B} \leftarrow \sum_{t=1}^\tau \tilde{\phi}_{1:B,t}$ (cf. Equation (11))
12: **return** Sketched per-example gradients $\tilde{\mathbf{g}}_{1:B} \in \mathbb{R}^{B \times m}$ and sketched step-resolved gradients $\{\tilde{\phi}_{1:B,t}\}_{t=1}^\tau$.

---

**Error bounds** Our sketched SDI estimator is unbiased and has variance that scales as $\mathcal{O}(1/m)$ in the sketch dimension $m$, corresponding to a typical estimation error that scales as $\mathcal{O}(1/\sqrt{m})$. We show this by specialising and tightening the TensorSketch variance bound of Pham & Pagh (2025,

Theorem 9):

**Lemma 1.** Fix examples $z, z'$, their SDI estimates $\mathcal{I}_t, \tilde{\mathcal{I}}_t$ and checkpoints $\mathcal{W} = \{\mathbf{w}_k\}_{k=1}^K$. For each $k$, define the training-gradient vector $\mathbf{g}_k := \nabla_{\mathbf{w}_{\text{body}}}\ell(\mathbf{w}_k; z)$, and for each $t \in \{1, \ldots, \tau\}$ define the test-step vector $\mathbf{p}_{k,t} := \phi_t(\mathbf{w}_k; z')$. Let $\mathcal{S}$ be the sketch map described above with sketch dimension parameter $m$, which we assume is even. Define the full and sketched dot products $\iota_{k,t} := \mathbf{g}_k \cdot \mathbf{p}_{k,t}$ and $\tilde{\iota}_{k,t} := \tilde{\mathbf{g}_k} \cdot \tilde{\mathbf{p}_{k,t}}$. Then, $\mathbb{E}[\tilde{\iota}_{k,t}] = \iota_{k,t}$ and $\text{Var}(\tilde{\iota}_{k,t}) \leq (4/m^2 + 6/m)\|\mathbf{g}_k\|_2^2\|\mathbf{p}_{k,t}\|_2^2$. Consequently, for each step $t$,

$$\mathbb{E}\left[\left(\tilde{\mathcal{I}}_t - \mathcal{I}_t\right)^2\right] \leq \left(\frac{4}{m^2} + \frac{6}{m}\right)\left(\sum_{k=1}^K \eta_k\|\mathbf{g}_k\|_2\|\mathbf{p}_{k,t}\|_2\right)^2. \tag{12}$$

*Proof.* See Appendix C. $\qquad\square$

We remark that the $(4/m^2 + 6/m)$ factor above comes from our novel bound of $\text{Var}(\tilde{\iota}_{k,t})$ and is *strictly tighter* than the $8/m$ factor derived in Pham & Pagh (2025). Moreover, our bound *cannot be improved in general:* there exists gradients $\mathbf{g}_k, \mathbf{p}_{k,t}$ such that the bound on $\text{Var}(\tilde{\iota}_{k,t})$ is tight. See Lemma 4 in the Appendix for the proof. Lemma 1 highlights the key scaling behaviour we need for SDI: the sketch error decays as $m$ grows and has *no explicit dependence on the parameter dimension* $|\mathbf{w}_{\text{body}}|$.

## 4. Experiments

**Scalability and Correctness** We first empirically validate that SDI is practical for larger-scale looped transformers. For this purpose, we instantiate a looped 135.1M-parameter GPT (Radford et al., 2019; Karpathy, 2025) with loop horizon $\tau = 32$. This yields an effective depth of 132 blocks and is FLOP-equivalent to a 1B-parameter model at sequence length 128, and all computations are in 32-bit floating point precision. Our TensorSketch SDI implementation matches the full stepwise-gradient baseline with low distortion while substantially reducing GPU memory consumption and negligible runtime overhead: On a single 48GB NVidia RTX A6000 GPU, our implementation translates into an order-of-magnitude increase in feasible batch size ($4 \to 40$ at $m = 2048$). Fidelity improves predictably with sketch dimension: at $m = 2048$ (used throughout the rest of the experimental section), relative SDI and TracIn errors are $0.0388 \pm 0.0030$ and $0.0220 \pm 0.0052$ (average$\pm$ standard deviation over 10 independent trials). Moreover, $m$ exhibits the expected $1/\sqrt{m}$ scaling (log–log slope: $-0.489$, see Appendix Figure 5). Conservation holds to numerical precision: the direct sketch of the full gradient matches the sum of per-step sketches up to $\approx 10^{-7}$ absolute error. Critically, our sketched SDI estimator adds little runtime overhead: at batch size 40, computing SDI adds only $2.55 \pm 0.002$s (10

trials) per weight checkpoint compared to an inference-only forward pass. Taken together, these results show that SDI is a scalable tool that can be deployed on LLM-scale looped transformers even on modest hardware. Additional post-hoc scaling experiments in Appendix D.2 show that the relative sketch cost decreases with model width, from $+116\%$ at 77M parameters to $+18\%$ at 1.2B parameters at fixed $\tau=32$, because the TensorSketch update scales linearly in width for fixed $m$, whereas the transformer backward pass scales roughly quadratically.

**SDI as a MechInterp hypothesis generator** *Mechanistic Interpretability* (MechInterp) attempts to explain model behaviour in terms of internal algorithmic mechanisms (*circuits* (Elhage et al., 2021)) rather than input/output attribution. We next demonstrate the synergy between data attribution and MechInterp by leveraging SDI to uncover and validate a finite-state circuit inside a looped transformer trained on the parity task. Following Fan et al. (2024), we train a causal looped transformer with a single weight-tied decoder block and additive input injection. Further model details can be found Appendix D. For a sequence with $n$ random bits, the model loops $\tau=n+2$ times, but we read out and compute the loss on the hidden state captured at loop iteration $n$. We train with a length curriculum that increases the maximum number of digits from 2 to 21 during training, and evaluate on length-40 out-of-distribution inputs where the trained model achieves $100\%$ accuracy. To probe the latent computation, we compute SDI trajectories for the loop-body parameters and demonstrate SDI on a "probe" input $(0101\ldots)$ of length 40. As shown in Figure 2A, SDI exhibits a sawtooth-like oscillation with apparent period 4, a final peak at step $n$ after which it drops to zero (since later loop iterations are causally unused and shown only for demonstration). In parallel, we track the model's forward dynamics via the logit margin at the answer token across loop iterations, which displays the same period-4 structure with sharp margin increases lagging SDI peaks by one iteration (Figure 2A). These step-resolved signatures suggest that the answer-token hidden state is cycling among a small number of discrete latent states.

We validate this hypothesis by explicitly unrolling the recurrent computation on the probe and recording the answer-token hidden state $\mathbf{h}_t$ after each loop iteration. We find strong recurrence at period 4 with $\mathbb{E}\cos(\mathbf{h}_t, \mathbf{h}_{t+4})\approx 0.95$ and a clear 4-state cycle in PCA space (Figure 2B). Discretising $\{\mathbf{h}_t\}$ by $k$-means with $k=4$ yields an almost-deterministic transition graph with a single dominant cycle except for an initial transient self-loop: $\widehat{P}=\begin{pmatrix} 0 & 0 & 1 & 0 \\ 0 & 0.09 & 0 & 0.91 \\ 0 & 1 & 0 & 0 \\ 1 & 0 & 0 & 0 \end{pmatrix}$, where $\widehat{P}_{ij}$ is the empirical probability of transitioning from state $i$ to $j$ at the next loop iteration (Figure 2B). Finally, we turn this circuit hypothesis into a proxy by replacing the model's linear readout by a discrete state assignment

(nearest $k$-means centroid at the readout step) followed by a state-to-parity lookup table learned on a small calibration set of alternating inputs. This proxy achieves $100\%$ accuracy on held-out alternating inputs across a wide range of lengths, including lengths beyond the training curriculum, and it remains *highly predictive on the actual length-40 test set performance of the model* ($\approx 93\%$ accuracy without refitting the discretisation). In summary, SDI provides a *visual, step-resolved* signature of latent periodic computation that guides a concrete mechanistic analysis and yields an interpretable proxy circuit. Taken together with the results of Fan et al. (2024), these findings indicate that looped transformer latent reasoning can implement discrete, length-generalisable finite-state algorithms for specific types of problems and inputs.

**Scaling laws of loop compute** A central promise of looped transformers is *test-time compute scaling* in activation space: at inference, one can allocate additional compute by unrolling the shared block for more recurrent iterations, without changing the input/output token length (in contrast to token-space test-time scaling). We study this regime on the Sudoku dataset from Wang et al. (2019). Following the approaches of Yang et al. (2023b); McLeish et al. (2025), we train a looped transformer with bidirectional attention, a single weight-tied decoder block and no input injection with a Poisson log-normal distribution over loop horizons with mean $\tau=32$ (clipped at $1\leq\tau\leq 64$), and evaluate the trained model at fixed test-time loop budgets between 4 and 64. Further model details can be found Appendix D. As shown in Figure 3A, harder puzzles are substantially more compute-sensitive: reducing loops sharply degrades accuracy, while increasing loops yields gains up to about $\tau\approx 64$ before saturating. To connect these accuracy scaling curves to the latent computation, we compute SDI and summarise the per-step magnitude of stepwise SDI scores as an *SDI energy* curve. Figure 3B shows that hard instances maintain higher SDI energy deeper into the recurrence (slower decay—note the logarithmic $y$-axis scaling), indicating that *later recurrent steps remain relevant when additional loop compute yields the largest gains*. In this sense, SDI complements accuracy-based scaling curves by characterising *where/how long* along the recurrent computation additional test-time compute remains salient.

**(Step-resolved) influence across difficulty** An advantage of algorithmic reasoning datasets is that they offer a precise notion of instance difficulty while avoiding the long-tail heterogeneity of natural text corpora (Feldman, 2020; Feldman & Zhang, 2020). For Sudoku, difficulty is the number of missing cells, yet all examples share a bounded, highly regular support (a fixed $9\times 9$ grid with the same local constraints), making this a clean testbed to study memorisation and influence without confounding from rare lexical patterns. Using (TracIn) self-influence as a proxy for memo-

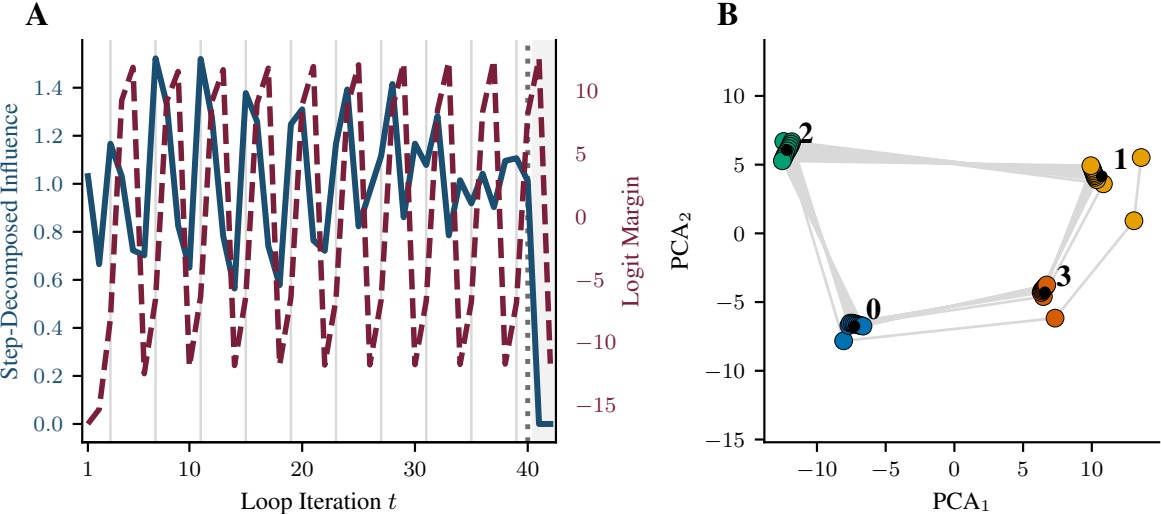

*Figure 2.* **SDI as a mechanistic discovery tool.** We train a looped transformer on parity and apply SDI to an alternating probe ($0101\dots$). **(A)** SDI and Logit Margin at the answer token across loop iterations $t$; light gray guides indicate the SDI peak phase (period 4) and the dotted gray line at $t{=}40$ marks the model's readout step. **(B)** PCA phase portrait of the answer-token hidden state across loop iterations, coloured by a $k{=}4$ discretization, revealing a four-state limit cycle.

risation as proposed by Feldman & Zhang (2020), we find that harder training puzzles are *systematically more memorised*: self-influence increases moderately with the number of blanks (Spearman $\rho{=}0.428$, $p{<}0.001$), and hard training puzzles (missing$\geq 45$) have roughly $2\times$ larger median self-influence than easy ones (missing$\leq 42$; Table 1). This mirrors the observations of Feldman & Zhang (2020) on natural data that harder examples tend to be memorised more, but interestingly, here the effect arises despite the absence of a long-tail support shift, suggesting an optimisation-driven bias tied to instance difficulty rather than rarity.

We next use SDI to examine how memorisation relates to generalisation (cross-influence) and *when* influence acts in the recurrent computation. Define the cross-influence mass of a training puzzle $z$ as $C(z){=}\sum_{z'\in\mathcal{D}_{\text{test}}}|\operatorname{TracIn}(z,z')|$. We find $C(z)$ increases moderately with training difficulty (Spearman $\rho{=}0.500$, $p{<}0.001$) and very strongly with self-influence ($\rho{=}0.919$, $p{<}0.001$); in particular, in this run we find *no* examples in the bottom quartile of self-influence that lie in the top quartile of $C(z)$, indicating that the most broadly cross-influential Sudoku puzzles are also the most memorised. Finally, SDI reveals a step-resolved signature of this coupling: decomposing the test-side SDI energy (Figure 3B) by training difficulty shows that hard training puzzles not only account for a larger share of $|\operatorname{TracIn}|$ mass, but also *place significantly more SDI energy in late loop iterations* (steps 17–32 of $\tau{=}32$; Table 1). Intuitively, hard puzzles are more memorised and help harder test puzzles more. This influence is exerted for longer throughout the loop. We view these results as closely related to the findings of Baldock et al. (2021) in (non-transformer) classification

NNs, who establish that harder examples are correctly predicted in deeper layers of the model and that these deeper layers (in our depth-recurrent transformers: later loop steps) are responsible for memorisation.

*Table 1.* **Influence, memorisation, and SDI timing on Sudoku.** *Easy train*: missing$\leq 42$; *Hard train*: missing$\geq 45$. On test bins (46–47 blanks = easy test; 49–50 blanks = hard test), *Share* is the mean fraction of per-test $|\operatorname{TracIn}|$ mass attributable to the given training group. *SDI late mass* is the mean fraction of SDI energy in late steps $t\geq 17$ (of $\tau{=}32$). $p$-values are Welch (train-train) or paired (train-test) t-tests and $p{<}0.001$ for all results in the table.

| | Easy train | Hard train |
|---|---|---|
| Self TracIn (median [IQR]) | 0.225 [0.155, 0.367] | 0.451 [0.303, 0.787] |
| Cross mass $C(z)$ (median [IQR]) | 3.07 [2.60, 3.82] | 4.43 [3.77, 5.80] |
| *On easy test puzzles (46–47 blanks):* | | |
| Share of $|\operatorname{TracIn}|$ mass | 28.9% | 37.7% |
| SDI late mass (steps 17–32) | 24.0% | 24.9% |
| *On hard test puzzles (49–50 blanks):* | | |
| Share of $|\operatorname{TracIn}|$ mass | 27.9% | 38.9% |
| SDI late mass (steps 17–32) | 24.3% | 25.3% |

**Counterfactual validation and signal cancellation** To test whether late-step SDI identifies training data with a distinct functional role, we performed a counterfactual data-removal experiment on Sudoku. We ranked training puzzles by their late-step SDI mass on a held-out validation split, removed the top $30\%$ of late-step influential examples, re-trained the model, selected checkpoints on validation data, and report final numbers on a disjoint hard-puzzle test split. As shown in Table 2, removing late-step influential data has a targeted effect: overall and easy-puzzle accuracy decrease only modestly, while hard-puzzle accuracy at $\tau{=}32$ drops significantly ($p{=}0.003$, $t$-test over 8 runs). This intervention

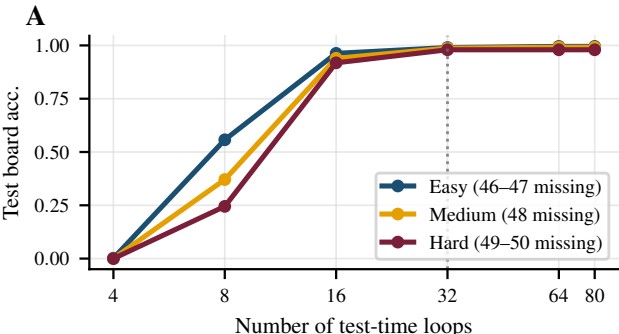

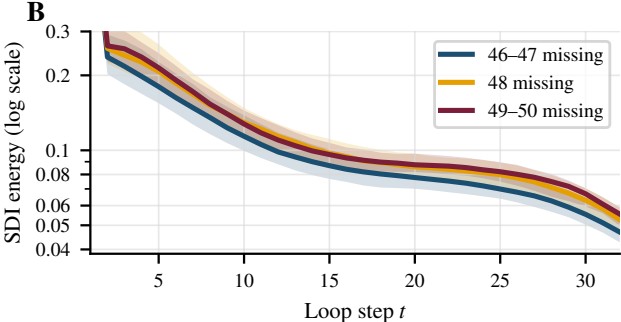

**Figure 3. Scaling laws of loop compute on SATNet Sudoku.**
**(A)** Test board accuracy (a board is correct iff all blank cells are correct) versus the number of test-time loops, stratified by puzzle difficulty (binned by the number of initial missing cells; more missing implies harder). Harder puzzles are substantially more compute-sensitive: reducing loops sharply degrades accuracy, while increasing loops yields gains up to about the training-mean depth $\tau \approx 32$ (dotted line) before saturating; we include >32 loops to show the plateau. **(B)** SDI energy across loop steps (median and IQR across puzzles in each difficulty bin), where SDI energy at step $t$ is the sum of absolute stepwise SDI scores across training points. Harder puzzles maintain higher SDI energy deeper into the recurrence (slower decay), mirroring their larger marginal gains from additional loop compute.

*Table 2.* **Counterfactual removal of late-step influential Sudoku data.** Training puzzles are ranked by late-step SDI mass on a held-out validation split and the top 30% are removed before retraining. Numbers are board accuracies in percent, mean±standard deviation over 8 runs. The hard-puzzle drop at $\tau$=32 is statistically significant ($p$=0.003).

| Condition | $\tau$=32 overall | $\tau$=64 overall | $\tau$=32, easy | $\tau$=32, hard |
|---|---|---|---|---|
| No removal | $98.6 \pm 0.7$ | $98.9 \pm 0.7$ | $98.7 \pm 0.6$ | $95.8 \pm 2.7$ |
| Remove top 30% late-step SDI | $97.8 \pm 0.7$ | $98.3 \pm 0.6$ | $98.0 \pm 0.8$ | $\mathbf{91.7 \pm 1.8}$ |

effect. Additional cancellation statistics for Sudoku are given in Appendix D.8.

**Nanochat**

Finally, we present a case study on *recursive Nanochat*, a 328.3M-parameter GPT-style chat model, retrofitting looped computation into Nanochat (Karpathy, 2025) as in McLeish et al. (2025). Details on the model training can be found Appendix D. In brief, the model is trained on a mixture of datasets (including reasoning datasets like GSM8K, ARC-Easy and ARC-Challenge and general language datasets like SmolTalk and SpellingBee) over three stages: pretraining, mid-training and supervised fine-tuning with truncated BPTT ($k$=4) and training loop steps sampled from a Poisson log-normal distribution with mean 4. We focus on GSM8K (train/test) as a reasoning dataset and evaluate SDI using various test-time recurrence counts $\tau \in \{2, \ldots, 16\}$ with sketch dimension $m$=2048 on the full answer. Importantly, we recompute SDI *without* truncation i.e., with full BPTT through all $\tau$ loop steps.

For all training stages that include GSM8K, influence is allocated almost exclusively to late loop steps: SDI increases nearly exponentially with $t$ with the last step alone usually contributing $\geq 50\%$ of the total influence. Figure 4 shows the normalised cumulative signed SDI sum per step $t$ for GSM8K exemplarily in the supervised fine-tuning stage.

Interestingly, the model seems to build up the entirety of its total influence within these last loop steps, *independently* of how many loop steps it is actually allowed to take. These results lead us to the hypothesis that the model implicitly encodes a "loop step counter" in the latent state: SDI reveals that—even without an explicit step embedding—the transformer seems able to represent progress through the loop because the model "knows what the four last loop steps are". This finding is corroborated by the task-specific performance on GSM8K plateauing for all test-time $\tau > 4$. We further corroborate this interpretation with two additional checks. First, the late-step concentration is robust across two additional seeds: for mid-training, the last-step SDI share is $0.532 \pm 0.010$ at $\tau$=8 and $0.506 \pm 0.017$ at $\tau$=16, while the last-four-step share is $0.885 \pm 0.006$ and $0.824 \pm 0.013$, respectively. For supervised fine-tuning, the last-step share is $0.604 \pm 0.026$ at $\tau$=8 and $0.606 \pm 0.028$ at $\tau$=16, while

is defined by *when* examples matter in the recurrence; scalar TracIn can rank examples by total influence, but cannot formulate a specifically late-step removal.

Signal cancellation provides a complementary example where SDI reveals structure that scalar TracIn collapses. In parity, we find a 20-bit training sequence and a 40-bit test sequence with near-zero cross-influence, TracIn =0.001, yet a large step-resolved magnitude, $\sum_t |\text{SDI}_t|$=17.24. The SDI trajectory is biphasic: the training sequence contributes positive cumulative influence during the first $\approx 19$ loop iterations ($+8.62$), then negative cumulative influence over the remaining $\approx 20$ iterations ($-8.62$), with the sign flip occurring around the training sequence length. Mechanistically, the training example teaches useful state transitions for the part of the recurrence corresponding to its own length, but misleads later steps where the test sequence extends beyond that support. Scalar TracIn therefore labels the pair as near-irrelevant, whereas SDI exposes a large opposing stepwise

the last-four-step share is $0.989 \pm 0.002$ and $0.987 \pm 0.001$. Second, a linear probe trained on activations at a fixed token position decodes the loop step within the TBPTT window with 98.56% held-out accuracy, whereas the same probe on a randomly initialised model with identical architecture is at chance (25.00%). This suggests that the model represents loop-step information in the latent state rather than relying only on token positional embeddings. We regard a deeper investigation of this phenomenon as a promising direction for future work.

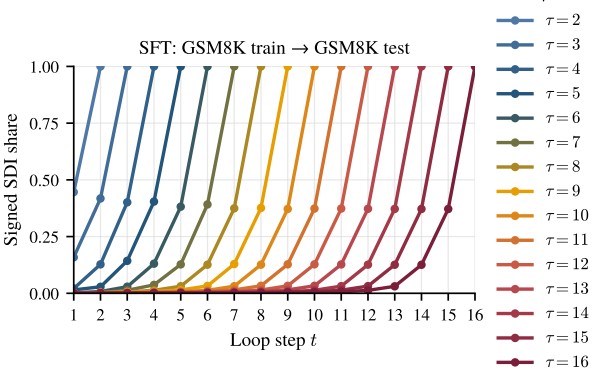

*Figure 4.* **Geometric influence growth across the loop horizon.** Signed SDI share per step $t$ for several analysis horizons $\tau$ in the supervised fine-tuning stage, recomputed with full BPTT through the recurrence from GSM8K train to GSM8K test. Removing truncation reveals a smooth, approximately geometric increase of stepwise influence with depth.

## 5. Discussion and Future Work

We have presented Step-Decomposed Influence (SDI), a framework that transforms data attribution from a scalar score into a temporal trajectory. By unrolling the gradient dynamics of looped transformers, SDI reveals not just *which* data points contribute to model performance, but *when* their influence manifests during the latent reasoning process.

**Limitations** TracIn admits the cleanest interpretation under (stochastic) gradient descent, where checkpoint weights $\eta_k$ directly reflect the learning-rate schedule. In modern training pipelines, the loop-body parameters may instead be updated by momentum, adaptive preconditioning, or optimizer-specific transformations. In such settings, our estimator should be viewed as a *TracIn-style gradient-similarity influence* computed along the training trajectory rather than a faithful approximation to an influence function derived from a specific optimization dynamics. A promising direction is to incorporate optimizer geometry more explicitly, e.g., by defining a preconditioned inner product. As an initial sanity check, we computed an AdamW-preconditioned variant on Sudoku, replacing each gradient inner product

by an elementwise second-moment preconditioned inner product. Despite a $131\times$ spread in effective per-coordinate learning rates, rankings were nearly unchanged: Spearman $\rho{=}0.979$ for TracIn, $\rho{=}0.934$ for SDI across steps, and 93.7% overlap among the top-5% most influential examples. This suggests that the unpreconditioned estimator is adequate for the ranking-based analyses in this work, while leaving optimizer-aware SDI as an important extension when absolute influence magnitudes are the target. Moreover, SDI is defined with respect to a particular unrolling of the computation graph and an analysis horizon $\tau$. When training uses randomized loop counts, adaptive halting, or truncated BPTT, step indices are not always comparable across examples, and truncation can systematically remove long-range credit assignment (early-step SDI is identically zero by construction). While we can recompute SDI with full BPTT for analysis (as in our Nanochat case study), scaling full-horizon SDI to very long recurrences remains expensive. Addressing this may require improved systems techniques, e.g., more aggressive activation recomputation/checkpointing or hardware scaling.

**Future Work** Like TracIn, SDI measures gradient alignment along a training trajectory; it does not on its own certify that removing or reweighting an example will change behaviour in a specific way, nor does it disentangle confounds such as shared features across many examples. Moreover, although sketching removes the need to materialize per-example gradients, computing dense train×test influence matrices is still costly at very large $|\mathcal{D}_{\text{train}}|$. A natural next step is to treat the sketched per-example (and per-step) vectors as an indexable embedding space: one can build approximate nearest-neighbour retrieval over $\widetilde{\nabla\ell(\cdot)}$ to find the most influential candidates for each query, and then refine SDI on a small retrieved set, enabling scalable workflows such as depth-targeted data curation and debugging.

Another promising direction is extending SDI beyond supervised next-token losses to modern alignment pipelines. For RLHF-style preference optimization, one could compute step-resolved influence of preference pairs on downstream behaviours, revealing whether alignment data primarily shapes early "instruction following" dynamics or late "reasoning/refinement" steps, and identifying training examples that disproportionately affect late-step computation where subtle failures often emerge. Similarly, in RL settings with verifiable rewards, SDI could help localize which trajectories and reward signals drive improvements at specific recurrent steps and diagnose step-local reward hacking. Finally, SDI suggests new ways to *use* the recurrence: step-energy curves and influence horizons could be turned into instance-wise stopping criteria or training-time regularisers that encourage useful computation to persist deeper into the loop, directly connecting interpretability signals to test-time compute allocation and model design.

## Acknowledgements

Martin J. Menten is funded by the German Research Foundation under project 532139938.

## Impact Statement

We introduce theoretical and practical tools for data attribution in looped transformers, bridging the gap between static influence scores and dynamic latent reasoning. While we acknowledge that precise influence estimation could theoretically be exploited to identify vulnerabilities for targeted data poisoning or to manipulate specific reasoning steps, we anticipate that these methods will be critical for improving model safety and transparency. By enabling practitioners to audit internal computations and trace the origins of model behaviours to specific training examples, our framework facilitates model debugging and dataset curation, ultimately supporting the development of more reliable and accountable AI reasoning systems.

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

# A. Extended Related Work

**Looped transformers** The concept of reusing transformer layers to increase computational depth without increasing parameters dates back to the Universal Transformer (Dehghani et al., 2018). Recent work has demonstrated that looped transformers can act as programmable computers capable of executing iterative algorithms (Giannou et al., 2023) and are superior at learning algorithms compared to standard transformers (Yang et al., 2023a). Saunshi et al. (2025) further bridge the gap to reasoning, arguing that looped models implicitly simulate Chain-of-Thought in continuous space while Gong et al. (2025) study formal properties of their learning dynamics. Scaling properties of these models have been investigated by Zhu et al. (2025b) and Fan et al. (2024), who show that loop-based architectures scale favourably for length generalisation in algorithmic tasks. Recent architectures such as Ouro (Zhu et al., 2025b) and RecurrentGemma (Botev et al., 2024) have successfully scaled these principles to LLM benchmarks. Our work complements this literature by providing the first (to our knowledge) step-resolved training-data influence estimator for weight-tied looped transformers, decomposing data influence across recurrent computation iterations rather than producing a scalar score aggregated over loop steps.

**Stepwise attribution in recurrent models** A broad literature studies stepwise attribution in recurrent non-transformer models, typically explaining a prediction by assigning importance to input timesteps (or tokens) rather than to internal recurrent computation iterations; examples include additive decompositions such as REAT (Du et al., 2019) and perturbation-based explainers such as TimeSHAP (Bento et al., 2021). Other work adapts training-data influence techniques to sequential settings, e.g., blockwise deletion for temporally correlated sequences (Alaa & Van Der Schaar, 2020) or influence for sequence tagging segments (Jain et al., 2022). These methods are complementary to SDI: we focus on weight-tied looped transformers and decompose training-example influence across the model's internal loop iterations, yielding a time-resolved influence trajectory over latent computation steps.

**Latent reasoning** Moving reasoning from explicit tokens to latent space is a rapidly growing frontier (Zhu et al., 2025a; Chen et al., 2025). Approaches like Coconut (Hao et al., 2024) train models to reason in continuous latent space, while others explore specialised architectures for recursive reasoning, such as block-recurrent dynamics in vision (Jacobs et al., 2025), Tiny Recursive Models (Jolicoeur-Martineau, 2025) and the very recent Universal Reasoning Model (Gao et al., 2025) that explicitly cites the recurrent design, inherited from Universal Transformers, as a key to SOTA results on very challenging reasoning benchmarks. Geiping et al. (2025) and McLeish et al. (2025) demonstrate that recurrent depth is a key enabler for scaling test-time compute. The recent work of Bogdan et al. (2025) studies the question of "which reasoning steps matter?" in reasoning models, but limits itself to models that reason in token space. On the other hand, the opacity of latent space in latent reasoning models remains a significant hurdle; our work attempts to bridge the gap by linking discrete latent computational steps back to the training data.

**Data attribution** Understanding model behaviour by identifying influential training examples is a fundamental goal of interpretability. The recent survey by Hammoudeh & Lowd (2024) highlights the utility of data attribution methods for debugging and data curation, while the seminal works of Feldman (2020); Feldman & Zhang (2020) have linked data attribution (and specifically influence estimation) to generalisation properties of deep neural networks. Classic approaches for influence estimation rely on Influence Functions (Koh & Liang, 2017), which estimate the effect of upweighting a training point via the Hessian. However, for modern deep networks and LLM-scale models, explicitly forming or inverting the Hessian is typically computationally prohibitive; practical influence-function implementations therefore rely on approximations. Gradient-based attribution methods, such as TracIn (Pruthi et al., 2020), approximate influence by tracing the dot product of gradients throughout training, a technique that has been theoretically connected to influence-function-style quantities under specific assumptions on the optimisation dynamics (Yang et al., 2024). We rely on TracIn over Influence Functions for two reasons: (1) Influence Functions typically assume a converged model whereas TracIn operates on the optimisation trajectory, allowing us to attribute behaviour to specific training dynamics captured at any intermediate checkpoint; (2) TracIn admits a clean linear decomposition over the recurrent computation as shown below, directly enabling the unrolled attribution of SDI, while it is unclear how to derive a similarly interpretable decomposition for curvature-based influence estimates involving the inverse Hessian across recurrent steps. Our work is complementary to current interpretability work that attributes model behaviour to *internal computation* at varying granularities (e.g., layer-wise, or circuit-level mechanisms (Elhage et al., 2021)) but these analyses typically assume a feedforward depth axis with untied parameters. In weight-tied looped transformers, repeated reuse of the same parameters collapses this axis, motivating attribution methods that explicitly resolve influence across recurrent *iterations* of a shared computation.

**Sketching and CountSketch/TensorSketch** Randomised sketching methods provide structured, memory-efficient alternatives to dense random projections for approximately preserving inner products. In particular, CountSketch constructs sparse

random linear maps that preserve dot products in expectation while being computationally cheap (Charikar et al., 2002). TensorSketch extends this idea to tensor products, enabling fast sketching of outer products via FFT-based convolution (Pagh, 2013; Pham & Pagh, 2025). Its core utility is the compression of sums of outer products without explicitly forming the high-dimensional tensors. We exploit this in SDI by observing that per-example gradients of transformer linear weights factor as outer products of activations and backpropagated signals; TensorSketch therefore allows us to compute per-example and per-step SDI features directly during backpropagation, avoiding materialisation of full per-example gradients. This is a key innovation over previous applications of TracIn that used dense (Pruthi et al., 2020) or sparse (Hu et al., 2025) Johnson Lindenstrauss (JL) projections (Johnson et al., 1984; Achlioptas, 2003). We elaborate on CountSketch and TensorSketch in the next subsection.

## B. Overview of CountSketch and TensorSketch

CountSketch was introduced in the context of estimating the frequency of items in a data stream, and TensorSketch in the context of approximating polynomial kernels in high dimensions. In this work, we instead use these techniques to compute SDI estimates. As a result, we use only a small subset of the results from the seminal papers of Pagh (2013); Pham & Pagh (2025). We provide an overview of the relevant results in this section, starting with some notation:

**Notation** Bolded lowercase letters such as $\mathbf{u}$ denote vectors and bolded uppercase letters such as $\mathbf{U}$ denote matrices. Let $[n] = \{1, \ldots, n\}$, $\mathbf{u} \otimes \mathbf{v} = \mathbf{u}\mathbf{v}^\top$ denote the outer product of $\mathbf{u}$ with $\mathbf{v}$, and $\langle \mathbf{u}, \mathbf{v} \rangle = \mathbf{u}^\top \mathbf{v}$ denote the vector inner product. The inner product between two matrices is the Frobenius inner product $\langle \mathbf{U}, \mathbf{V} \rangle = \text{tr}(\mathbf{U}^\top \mathbf{V})$. Note that the Frobenius inner product is the same as flattening the matrices into 2 vectors, then computing the vector inner product. In the main body, we flattened all matrices into vectors, and used the notation $\mathbf{u} \cdot \mathbf{v}$ denote the inner product for both vectors and matrices. In this Appendix, we keep the shapes of the matrices (we do not flatten), and use the notation $\langle \cdot, \cdot \rangle$ to denote the inner product for both vectors and matrices. Let $\circ$ denote the Hadamard product so that $\mathbf{Y} = \mathbf{U} \circ \mathbf{V}$ means $\mathbf{Y}_{ij} = \mathbf{U}_{ij}\mathbf{V}_{ij}$.

Now, we introduce the concept of $k$-wise independence, which is needed for reasoning about the randomness in hash functions:

**Definition 2.** A family $\mathcal{H}$ of functions $f : [d] \to [m]$ is k-wise independent if for any set of integers $\{i_1, \ldots, i_k\} \subseteq [d]$, and a random $f \in \mathcal{H}$, the vector of hash values $\{f(i_1), \ldots, f(i_k)\}$ is uniform in $[m]^k$.

### B.1. CountSketch

**Definition 3** (Pham & Pagh (2025)). Given $h : [d] \to [m]$ sampled from a 2-wise independent family, and $s : [d] \to \{-1, 1\}$ sampled from a 4-wise independent family, a CountSketch CS of a vector $\mathbf{x} \in \mathbb{R}^d$ (often denoted $\tilde{\mathbf{x}}$ and referred to as the sketch of $\mathbf{x}$) is defined as the linear map

$$\mathsf{CS}(\mathbf{x})_j = \sum_{i \in [d]: h(i) = j} s(i)\mathbf{x}_i$$
$$= \sum_{i \in [d]} \mathbb{1}[h(i) = j]s(i)\mathbf{x}_i$$

In our settings, $m < d$. Therefore, for fixed hash functions $(h, s)$ applied to real-valued vectors, CountSketch can be used as an efficient and random low-dimensional embedding for high dimensional vectors. One may worry that reusing the same hash functions for all vectors correlates the embeddings. This is a valid concern, and the purpose of the following lemma, due to Pham & Pagh (2025), is to show that these correlations can be controlled.

**Lemma 2** (Pham & Pagh (2025)). Given $h : [d] \to [m]$ sampled from a 2-wise independent family and $s : [d] \to \{-1, 1\}$ sampled from a 4-wise independent family, denote by $\tilde{\mathbf{x}}, \tilde{\mathbf{y}} \in \mathbb{R}^m$ the respective CountSketches of vectors $\mathbf{x}, \mathbf{y} \in \mathbb{R}^d$ based on the hash functions $h, s$. We have:

$$\mathbb{E}[\langle \tilde{\mathbf{x}}, \tilde{\mathbf{y}} \rangle] = \langle \mathbf{x}, \mathbf{y} \rangle$$

$$\text{Var}[\langle \tilde{\mathbf{x}}, \tilde{\mathbf{y}} \rangle] = \frac{1}{m}\left( \sum_{i \neq j \in [d]} \mathbf{x}_i^2 \mathbf{y}_j^2 + \sum_{i \neq j \in [d]} \mathbf{x}_i \mathbf{y}_i \mathbf{x}_j \mathbf{y}_j \right) \leq \frac{2}{m}\|\mathbf{x}\|_2^2\|\mathbf{y}\|_2^2$$

*Proof.* See Lemma 6 of Pham & Pagh (2025). $\qquad\square$

This means that the (random) CountSketch embedding space approximately conserves inner products between vectors, and the variance induced by the hashes increases as $1/m$.

**Remark 1.** The function CS is linear. That is, the CountSketch (using hashes $(h, s)$) of the sum of $n$ vectors is the same as the sum of CountSketches, where each CountSketch uses the same hashes $(h, s)$. While obvious from the definition, we highlight this insight since the sum of CountSketches, each using the same hash, is the sum of $n$ correlated vectors. That the variance bound in Lemma 2 holds in this case is central to the usefulness of CountSketch.

### B.2. TensorSketch

TensorSketch extends the concept of CountSketch to outer products and more generally to $p$-th tensor powers. [1] For our purposes, we will only make use of their applications to outer products. TensorSketch is a random embedding that takes as input two vectors $\mathbf{x} \in \mathbb{R}^d, \mathbf{x}' \in \mathbb{R}^{d'}$, and outputs a sketch of $\mathbf{X} = \mathbf{x} \otimes \mathbf{x}'$ (denoted $\tilde{\mathbf{X}}$). The most naive way to do this is to apply two independent CountSketches to the inputs to get $\tilde{\mathbf{x}} = \mathsf{CS}_1(\mathbf{x}), \tilde{\mathbf{x}}' = \mathsf{CS}_2(\mathbf{x}')$, then outputting $\tilde{\mathbf{X}} = \tilde{\mathbf{x}} \otimes \tilde{\mathbf{x}}'$. This yields a sketch of shape $m^2$ assuming the sketches $\tilde{\mathbf{x}}, \tilde{\mathbf{x}}' \in \mathbb{R}^m$. This is *not* what TensorSketch does. Like the naive approach, it does make use of two independent CountSketches, but in contrast to generating an outer product sketch of size $m^2$, it instead exploits the structure of hashes to generate an outer product sketch of size $m$. Formally:

**Definition 4** ((Pham & Pagh, 2025)). Let $h_1 : [d] \to [m], h_2 : [d'] \to [m]$ be two hashes sampled independently from a 2-wise independent family. Let $s_1 : [d] \to \{-1, 1\}, s_2 : [d'] \to \{-1, 1\}$ be two hashes sampled independently from a 4-wise independent family. Define two new hashes $H, S$ via:

$$H(i, j) = h_1(i) + h_2(j) \mod m$$
$$S(i, j) = s_1(i)s_2(j).$$

A TensorSketch TS of vectors $\mathbf{x} \in \mathbb{R}^d, \mathbf{x}' \in \mathbb{R}^{d'}$ is defined as the map $\mathsf{TS} : \mathbb{R}^d \times \mathbb{R}^{d'} \to \mathbb{R}^m$ such that:

$$\mathsf{TS}(\mathbf{x}, \mathbf{x}')_\alpha = \sum_{i \in [d], j \in [d']} \mathbb{1}[H(i, j) = \alpha] S(i, j) \mathbf{x}_i \mathbf{x}'_j.$$

Moreover, let $\mathsf{CS}_1$ denote the CountSketch generated by $(h_1, s_1)$ and let $\tilde{\mathbf{x}} = \mathsf{CS}_1(\mathbf{x})$, and similarly for $\mathbf{x}'$. The TensorSketch $\mathsf{TS}(\mathbf{x}, \mathbf{x}')$ can be computed efficiently from $\tilde{\mathbf{x}}, \tilde{\mathbf{x}}'$ via circular convolution:

$$\mathsf{TS}(\mathbf{x}, \mathbf{x}')_\alpha = \sum_{k \in [m]} \mathbf{x}_k \mathbf{x}'_{1+(\alpha-k) \mod m}$$
$$= \mathsf{iFFT}(\mathsf{FFT}(\tilde{\mathbf{x}}) \circ \mathsf{FFT}(\tilde{\mathbf{x}}')).$$

After introducing TensorSketches as above, Pham & Pagh (2025) focus on using TensorSketch as low-dimensional embeddings for the $p$-th tensor product of a vector $\mathbf{x}$. Their error analysis assumes that given two vectors $(\mathbf{x}, \mathbf{y}) \in \mathbb{R}^d$, one is interested in finding a representation of $\mathbf{x}^{(p)}, \mathbf{y}^{(p)} \in \mathbb{R}^{d^p}$ that preserves the inner product $\langle \mathbf{x}^{(p)}, \mathbf{y}^{(p)} \rangle = \langle \mathbf{x}, \mathbf{y} \rangle^p$ in expectation and with low variance. We take a different approach, and view TensorSketch as a low-dimensional embedding for matrices in $\mathbb{R}^{d \times d'}$. We do this by extending the scope of the definition of TensorSketch in Definition 4, which initially defined TS as taking as input two vectors $\mathbf{x}, \mathbf{x}'$, to a definition that takes as input a matrix.

**Lemma 3** (TensorSketch as a Linear Operator). TensorSketch TS induces a linear operator $\mathcal{T} : \mathbb{R}^{d \times d'} \to \mathbb{R}^m$ on the space of matrices. Explicitly, for any matrix $\mathbf{A} \in \mathbb{R}^{d \times d'}$, we define the sketch $\mathcal{T}(\mathbf{A})$ by:

$$\mathcal{T}(\mathbf{A})_\alpha = \sum_{\substack{i \in [d] \\ j \in [d']}} \mathbb{1}[H(i, j) = \alpha] S(i, j) \mathbf{A}_{ij},$$

where $H, S$ are the same as in Definition 4. For general matrices, there is no structure to exploit and therefore no convolution to make the computation of $\mathcal{T}(\mathbf{A})$ efficient. However, if a matrix $\mathbf{X}$ is a sum of outer products $\mathbf{X} = \sum_{k=1}^K \mathbf{x}^{(k)} \otimes \mathbf{x}'^{(k)})$, then the sketch satisfies the property:

$$\mathcal{T}(\mathbf{X}) = \sum_{k=1}^K \mathsf{TS}(\mathbf{x}^{(k)}, \mathbf{x}'^{(k)}).$$

---

[1] The $p$-th tensor power of $x$ is $x \underbrace{\otimes \cdots \otimes}_{p \text{ times}} x$

Therefore, $\mathcal{T}$ when applied to this particular matrix structure can be efficiently computed using the convolutional algorithm used in TS and defined in Definition 4.

*Proof.* Let $C_{i,j}^{\alpha} = \mathbb{1}[H(i,j) = \alpha]S(i,j)$. By definition, the $\alpha$-th component of the sketch of $\mathbf{X}$ is:

$$
\begin{aligned}
\mathcal{T}(\mathbf{X})_\alpha &= \sum_{i\in[d],j\in[d']} C_{i,j}^{\alpha} \mathbf{X}_{ij} \\
&= \sum_{i\in[d],j\in[d']} C_{i,j}^{\alpha} \left( \sum_{k=1}^{K} \mathbf{x}_i^{(k)} (\mathbf{x}'^{(k)})_j \right) \\
&= \sum_{k=1}^{K} \left( \sum_{i\in[d],j\in[d']} C_{i,j}^{\alpha} \mathbf{x}_i^{(k)} (\mathbf{x}'^{(k)})_j \right) \\
&= \sum_{k=1}^{K} \mathsf{TS}(\mathbf{x}^{(k)}, \mathbf{x}'^{(k)})_\alpha.
\end{aligned}
$$

Thus, in the special case of sums of outer products, the sketch of the sum is the sum of the sketches. Notably, this means we can compute $\mathcal{T}(\mathbf{X})$ without materializing any of the $d \times d'$ matrices $\mathbf{x}^{(k)} \otimes \mathbf{x}'^{(k)}$. $\square$

With the above lemma, we can interpret TensorSketch (i.e. the linear operator $\mathcal{T}$) as a linear low-dimensional random embedding for matrices, though this embedding takes $\mathcal{O}(dd')$ time to compute for general matrices. Since gradients in sequence-to-sequence models are sums of outer-products however, the TensorSketch of gradients can be computed without materializing the outer-products utilizing convolutions, and is therefore efficient and scalable.

Just like the CountSketch case, one may worry that repeatedly using the same hash functions $(H, S)$ for various gradients correlates the embeddings, leading to a possibly unbounded error when computing inner products like what is needed in and TracIn. The following lemma shows that this is not the case.

**Lemma 4** (Error Analysis of the Linear Operator $\mathcal{T}$)**.** Given hashes $(H, S)$ as defined in Definition 4 and matrices $\mathbf{X}, \mathbf{Y} \in \mathbb{R}^{d \times d'}$, let $\tilde{\mathbf{X}} = \mathcal{T}(\mathbf{X}) \in \mathbb{R}^m, \tilde{\mathbf{Y}} = \mathcal{T}(\mathbf{Y}) \in \mathbb{R}^m$ denote the respective TensorSketches of matrices $\mathbf{X}, \mathbf{Y}$ as outlined in Lemma 3. We have:

$$
\mathbb{E}[\langle \tilde{\mathbf{X}}, \tilde{\mathbf{Y}} \rangle] = \langle \mathbf{X}, \mathbf{Y} \rangle
$$
$$
\mathrm{Var}[\langle \tilde{\mathbf{X}}, \tilde{\mathbf{Y}} \rangle] = \frac{2}{m^2}(P_1 - N_1) + \frac{1}{m}(P_2 - N_2)
$$
$$
\leq \|\mathbf{X}\|_F^2 \|\mathbf{Y}\|_F^2 \left( \frac{6}{m} + \frac{4}{m^2} \right)
$$

Where $P_1, N_1, P_2, N_2$ are defined as:

$$
\begin{aligned}
P_1 &= \left\| \mathbf{X}^T\mathbf{Y} \right\|_F^2 + \left\| \mathbf{X}\mathbf{Y}^T \right\|_F^2 + 2\langle \mathbf{X} \circ \mathbf{X}, \mathbf{Y} \circ \mathbf{Y} \rangle \\
N_1 &= \|\mathrm{diag}(\mathbf{X}^T\mathbf{Y})\|_2^2 + \|\mathrm{diag}(\mathbf{X}\mathbf{Y}^T)\|_2^2 + \mathrm{tr}((\mathbf{X}\mathbf{X}^T) \circ (\mathbf{Y}\mathbf{Y}^T)) + \mathrm{tr}((\mathbf{X}^T\mathbf{X}) \circ (\mathbf{Y}^T\mathbf{Y})) \\
P_2 &= P_1 + 2\mathrm{tr}((\mathbf{X}^T\mathbf{Y})^2) + \langle \mathbf{X}, \mathbf{Y} \rangle^2 + \|\mathbf{X}\|_F^2 \|\mathbf{Y}\|_F^2 \\
N_2 &= N_1 + 2\left[ \|\mathrm{diag}(\mathbf{X}^T\mathbf{Y})\|_2^2 + \|\mathrm{diag}(\mathbf{X}\mathbf{Y}^T)\|_2^2 \right]
\end{aligned}
$$

Moreover, the variance bound is tight in that there exists matrices $\mathbf{X}, \mathbf{Y}$ that get arbitrarily close to the upper-bound.

*Proof.* Starting from

$$
\tilde{\mathbf{X}}_\alpha = \sum_{\substack{i\in[d]\\j\in[d']}} \mathbb{1}[H(i,j) = \alpha]S(i,j)\mathbf{X}_{ij}, \quad \tilde{\mathbf{Y}}_\alpha = \sum_{\substack{k\in[d]\\\ell\in[d']}} \mathbb{1}[H(k,\ell) = \alpha]S(k,\ell)\mathbf{Y}_{k\ell}
$$

We introduce the notation $\mathbf{u} = (i, j)$ and $\mathbf{v} = (k, \ell)$ to simplify notation:

$$\tilde{\mathbf{X}}_\alpha = \sum_{\mathbf{u} \in [d] \times [d']} \mathbb{1}[H(\mathbf{u}) = \alpha] S(\mathbf{u}) \mathbf{X_u}, \quad \tilde{\mathbf{Y}}_\alpha = \sum_{\mathbf{v} \in [d] \times [d']} \mathbb{1}[H(\mathbf{v}) = \alpha] S(\mathbf{v}) \mathbf{Y_v}.$$

Following Pham & Pagh (2025), define $\xi_{\mathbf{u},\mathbf{v}} = \mathbb{1}[H(\mathbf{u}) = H(\mathbf{v})]$. Expanding out the inner product between the two sketches gives:

$$\begin{aligned}
\langle \tilde{\mathbf{X}}, \tilde{\mathbf{Y}} \rangle &= \sum_{\alpha \in [m]} \tilde{\mathbf{X}}_\alpha \tilde{\mathbf{Y}}_\alpha \\
&= \sum_{\alpha \in [m]} \sum_{\mathbf{u},\mathbf{v} \in [d] \times [d']} \mathbb{1}[H(\mathbf{u}) = \alpha] S(\mathbf{u}) \mathbf{X_u} \, \mathbb{1}[H(\mathbf{v}) = \alpha] S(\mathbf{v}) \mathbf{Y_v} \\
&= \sum_{\mathbf{u},\mathbf{v} \in [d] \times [d']} S(\mathbf{u}) S(\mathbf{v}) \mathbf{X_u} \mathbf{Y_v} \left( \sum_{\alpha \in [m]} \mathbb{1}[H(\mathbf{u}) = \alpha] \mathbb{1}[H(\mathbf{v}) = \alpha] \right) \\
&= \sum_{\mathbf{u},\mathbf{v} \in [d] \times [d']} S(\mathbf{u}) S(\mathbf{v}) \mathbf{X_u} \mathbf{Y_v} \, \mathbb{1}[H(\mathbf{u}) = H(\mathbf{v})] \\
&= \sum_{\mathbf{u},\mathbf{v} \in [d] \times [d']} S(\mathbf{u}) S(\mathbf{v}) \mathbf{X_u} \mathbf{Y_v} \xi_{\mathbf{u},\mathbf{v}} \\
&= \langle \mathbf{X}, \mathbf{Y} \rangle + \sum_{\mathbf{u} \neq \mathbf{v} \in [d] \times [d']} S(\mathbf{u}) S(\mathbf{v}) \mathbf{X_u} \mathbf{Y_v} \xi_{\mathbf{u},\mathbf{v}}
\end{aligned}$$

Since $s_1$ and $s_2$ are 2-wise independent, their product $S$. This means $S(\mathbf{u}) S(\mathbf{v})$ (if $\mathbf{u} \neq \mathbf{v}$) is the product of two independent Rademacher random variables, which has mean 0 in expectation. Therefore $\mathbb{E}[\langle \tilde{\mathbf{X}}, \tilde{\mathbf{Y}} \rangle] = \langle \mathbf{X}, \mathbf{Y} \rangle$.

Now, for the variance. We write:

$$\text{Var}[\langle \tilde{\mathbf{X}}, \tilde{\mathbf{Y}} \rangle] = \mathbb{E}[\langle \tilde{\mathbf{X}}, \tilde{\mathbf{Y}} \rangle^2] - \langle \mathbf{X}, \mathbf{Y} \rangle^2. \tag{13}$$

We expand $\langle \tilde{\mathbf{X}}, \tilde{\mathbf{Y}} \rangle^2$ to:

$$\begin{aligned}
\langle \tilde{\mathbf{X}}, \tilde{\mathbf{Y}} \rangle^2 &= \left( \langle \mathbf{X}, \mathbf{Y} \rangle + \sum_{\mathbf{u} \neq \mathbf{v} \in [d] \times [d']} S(\mathbf{u}) S(\mathbf{v}) \mathbf{X_u} \mathbf{Y_v} \xi_{\mathbf{u},\mathbf{v}} \right)^2 \\
&= \langle \mathbf{X}, \mathbf{Y} \rangle^2 + 2 \langle \mathbf{X}, \mathbf{Y} \rangle \sum_{\mathbf{u} \neq \mathbf{v} \in [d] \times [d']} S(\mathbf{u}) S(\mathbf{v}) \mathbf{X_u} \mathbf{Y_v} \xi_{\mathbf{u},\mathbf{v}} + \left( \sum_{\mathbf{u} \neq \mathbf{v} \in [d] \times [d']} S(\mathbf{u}) S(\mathbf{v}) \mathbf{X_u} \mathbf{Y_v} \xi_{\mathbf{u},\mathbf{v}} \right)^2. \tag{14}
\end{aligned}$$

The first term of Equation (14) is a constant and cancels the same term in Equation (13). The second term disappears when we take the expectation due to the 2-wise independence of $S$. The third term is non-zero since the 2-wise independence of $S$ introduces correlations. We are left with:

$$\begin{aligned}
\text{Var}[\langle \tilde{\mathbf{X}}, \tilde{\mathbf{Y}} \rangle] &= \mathbb{E}\left[ \left( \sum_{\mathbf{u} \neq \mathbf{v} \in [d] \times [d']} S(\mathbf{u}) S(\mathbf{v}) \mathbf{X_u} \mathbf{Y_v} \xi_{\mathbf{u},\mathbf{v}} \right)^2 \right] \\
&= \mathbb{E}\left[ \sum_{\substack{\mathbf{u} \neq \mathbf{v} \in [d] \times [d'] \\ \mathbf{u}' \neq \mathbf{v}' \in [d] \times [d']}} S(\mathbf{u}) S(\mathbf{v}) S(\mathbf{u}') S(\mathbf{v}') \mathbf{X_u} \mathbf{Y_v} \mathbf{X_{u'}} \mathbf{Y_{v'}} \xi_{\mathbf{u},\mathbf{v}} \xi_{\mathbf{u}',\mathbf{v}'} \right] \\
&= \sum_{\substack{\mathbf{u} \neq \mathbf{v} \in [d] \times [d'] \\ \mathbf{u}' \neq \mathbf{v}' \in [d] \times [d']}} \mathbb{E}\left[ S(\mathbf{u}) S(\mathbf{v}) S(\mathbf{u}') S(\mathbf{v}') \right] \mathbf{X_u} \mathbf{Y_v} \mathbf{X_{u'}} \mathbf{Y_{v'}} \mathbb{E}\left[ \xi_{\mathbf{u},\mathbf{v}} \xi_{\mathbf{u}',\mathbf{v}'} \right] \tag{15}
\end{aligned}$$

Where we used the linearity of expectation and the fact that randomness of $S$ is independent of the randomness of $H$ to split the expectation into two. Evaluating the expectation Equation (15) is non-trivial and will take multiple pages of work. We begin by splitting the expectation over $S$ into three expectations over $s_1, s_2$ and $H$(recall that $s_1, s_2$ are independent from one other and 4-wise independent, and $H$ is independent of $s_1, s_2$ and 2-wise independent). The full expression for the sum is (in the unrolled indices) is:

$$
\begin{aligned}
\text{Var}[\langle \tilde{\mathbf{X}}, \tilde{\mathbf{Y}} \rangle] = \sum_{\substack{i,i',k,k' \in [d] \\ j,j',\ell,\ell' \in [d']}} & \mathbb{E}[s_1(i)s_1(k)s_1(i')s_1(k')]\mathbb{E}[s_2(j)s_2(\ell)s_2(j')s_2(\ell')] \\
& \cdot \mathbb{1}[i \neq k \text{ or } j \neq \ell]\mathbb{1}[i' \neq k' \text{ or } j' \neq \ell')]\mathbb{E}\left[\xi_{ij,k\ell}\xi_{i'j',k'\ell'}\right] \\
& \cdot \mathbf{X}_{ij}\mathbf{Y}_{k\ell}\mathbf{X}_{i'j'}\mathbf{Y}_{k'\ell'}
\end{aligned}
\tag{16}
$$

where

$$
\mathbb{E}\left[\xi_{ij,k\ell}\xi_{i'j',k'\ell'}\right] = \Pr[(H(i,j) = H(k,\ell) \text{ and } (H(i',j') = H(k',\ell')]
$$

Note that the $s_1, s_2$ expectations can only be 0 or 1. Therefore, in order to determine what this sum evaluates to, we will first identify the indices where the expectations over $s_1, s_2$ are non-zero. Then, we will determine if the indicator functions rule out any of those indices. Then, we will find what the $\xi$ expectations equal under the remaining indices, then we will do the sums over the tensors $\mathbf{X}, \mathbf{Y}$.

In order for a term to contribute to the sum, both of the $s_1, s_2$ expectations must be 1. Lets focus on the expectation over $s_1$. The only way for this to be non-zero is if the 4 indices are all equal or if the 4 indices form 2 equal pairs. This is because if three indices are equal then we have $\mathbb{E}[s(i)^3 s(j)] = \mathbb{E}[s(i)^3]\mathbb{E}[s(j)] = 0$. The same thing happens if the 4 indices are not the same. For one expectation, this means we have the following cases:

$$
\mathbb{E}[s_1(i)s_1(k)s_1(i')s_1(k')] = \begin{cases} 1 & \text{if } i = k \text{ and } k' = i' \\ 1 & \text{if } i = i' \neq k = k' \\ 1 & \text{if } i = k' \neq k = i' \\ 0 & \text{otherwise} \end{cases}
$$

Note that the first case jointly covers when $i = k \neq k' = i'$ and $i = k = k' = i'$. We have the same 3 cases for the other expectation, which leads to nine total cases for when the total expectation is 1:

$$
\mathbb{E}[s_2(j)s_2(\ell)s_2(j')s_2(\ell')] = \begin{cases} 1 & \text{if } j = \ell \text{ and } \ell' = j' \\ 1 & \text{if } j = j' \neq \ell = \ell' \\ 1 & \text{if } j = \ell' \neq \ell = j' \\ 0 & \text{otherwise} \end{cases}
$$

Lets go through each of the 9 cases individually. The order we go in will appear random upon a first reading, but as the math proceeds it will become clear why we go in the order we do. In each case, we will first evaluate what the $\xi$ expectation simplifies to, then we will do the sum over the tensors $\mathbf{X}, \mathbf{Y}$, which we will also write into matrix notation. The following summation identities will be useful:

$$
\sum_{\substack{i \neq k \in [d] \\ j \neq \ell \in [d']}} A_{ijk\ell} = \sum_{\substack{i,k \in [d] \\ j,\ell \in [d']}} A_{ijk\ell} - \sum_{\substack{i \in [d] \\ j,\ell \in [d']}} A_{iji\ell} - \sum_{\substack{i,k \in [d] \\ j \in [d']}} A_{ijkj} + \sum_{\substack{i \in [d] \\ j \in [d']}} A_{ijij}
\tag{17}
$$

$$
\sum_{\substack{i \neq k \in [d] \\ j \neq \ell \in [d']}} A_{ijk\ell} = \sum_{\substack{i,k \in [d] \\ j \neq \ell \in [d']}} A_{ijk\ell} - \sum_{\substack{i \in [d] \\ j \neq \ell \in [d']}} A_{iji\ell}
\tag{18}
$$

The following matrix equalities will also be useful:

$$
(\mathbf{XY})_{ij} = \sum_k \mathbf{X}_{ik}\mathbf{Y}_{kj} \quad (\mathbf{X}^T\mathbf{Y})_{ij} = \sum_k \mathbf{X}_{ki}\mathbf{Y}_{kj} \quad (\mathbf{XY}^T)_{ij} = \sum_k \mathbf{X}_{ik}\mathbf{Y}_{jk}.
$$

We begin with the 9 cases:

0: $(i = k$ and $k' = i')$ and $(j = \ell$ and $\ell' = j')$.

While this case makes the expectation 1, terms with this indices in the sum Equation (16) are zero due to the indicator functions, so this case can be discarded.

1: $(i = k$ and $k' = i')$ and $(j = j' \neq \ell = \ell')$.

We begin by noting that the expectation over $\xi$ is the same for all elements in the sum:

$$\mathbb{E}\left[\xi_{ij,i\ell}\xi_{i'j,i'\ell}\right] = \Pr[(H(i,j) = H(i,\ell) \text{ and } (H(i',j) = H(i',\ell)]$$

Expanding out the events in the probability, we have:

$$h_1(i) + h_2(j) \equiv h_1(i) + h_2(\ell) \pmod{m} \tag{19}$$
$$h_1(i') + h_2(j) \equiv h_1(i') + h_2(\ell) \pmod{m}. \tag{20}$$

where here we use $\equiv$ to denote equality under the modulo operation. Remember that we are interested in the probability that both events Equation (19) and Equation (20) occur. Subtracting like terms, we have that the events simplify to:

$$h_2(j) - h_2(\ell) \equiv 0 \pmod{m}$$

Since $h_2$ is 2-wise independent and $j \neq \ell$, $h_2(j), h_2(\ell)$ can be treated as two independent uniform random variables in $[m]$, and therefore $h_2(j) - h_2(\ell) \bmod m$ is a uniform random variable in $\{0, 1, \ldots, m-1\}$. Therefore the probability that it equals 0 is simply $1/m$, and the expectation over $\xi$ is:

$$\mathbb{E}\left[\xi_{ij,i\ell}\xi_{i'j,i'\ell}\right] = \frac{1}{m}$$

With the $\xi$ expectation handled, we move on to the sum over the tensors $\mathbf{X}, \mathbf{Y}$. The sum becomes:

$$
\begin{aligned}
T_1 &= \sum_{\substack{i,i' \in [d] \\ j \neq \ell \in [d']}} \mathbf{X}_{ij}\mathbf{Y}_{i\ell}\mathbf{X}_{i'j}\mathbf{Y}_{i'\ell} \\
&= \sum_{j \neq \ell} \left(\sum_{i \in [d]} \mathbf{X}_{ij}\mathbf{Y}_{i\ell}\right)\left(\sum_{i' \in [d]} \mathbf{X}_{i'j}\mathbf{Y}_{i'\ell}\right) \\
&= \sum_{j \neq \ell}(\mathbf{X}^T\mathbf{Y})_{j\ell}(\mathbf{X}^T\mathbf{Y})_{j\ell} \\
&= \left\|\mathbf{X}^T\mathbf{Y}\right\|_F^2 - \sum_{j \in [d']}(\mathbf{X}^T\mathbf{Y})_{jj}(\mathbf{X}^T\mathbf{Y})_{jj} \\
&= \left\|\mathbf{X}^T\mathbf{Y}\right\|_F^2 - \left\|\operatorname{diag}(\mathbf{X}^T\mathbf{Y})\right\|_2^2
\end{aligned}
$$

Note that the actual contribution to the sum is $\frac{1}{m}T_1$. We exclude the $1/m$ term in the derivation above for notational clarity, and also because in the upcoming cases, the same $T_1$ term appears although the value of the $\xi$ expectation is different.

2: $(i = k' \neq k = i')$ and $(j = j' \neq \ell = \ell')$

Similar to $T_1$, here the expectation over $\xi$ is the same for all elements in the sum:

$$\mathbb{E}\left[\xi_{ij,k\ell}\xi_{kj,i\ell}\right] = \Pr[(H(i,j) = H(k,\ell) \text{ and } (H(k,j) = H(i,\ell)]$$

Expanding out the events in the probability, we have:

$$h_1(i) + h_2(j) \equiv h_1(k) + h_2(\ell) \pmod{m}$$
$$h_1(k) + h_2(j) \equiv h_1(i) + h_2(\ell) \pmod{m}.$$

We can rearrange the events as:

$$h_1(i) - h_1(k) \equiv -(h_2(j) - h_2(\ell)) \pmod{m}$$
$$h_1(i) - h_1(k) \equiv h_2(j) - h_2(\ell) \pmod{m}.$$

Since $h_1, h_2$ are independent, the event $A = h_1(i) - h_1(k) \bmod m$ is independent of $B = h_2(j) - h_2(\ell) \bmod m$. Moreover, $A, B$ are uniform random variables in $\{0, 1, 2, \ldots, m-1\}$. Our equations of interest simplify to:

$$A + B \equiv 0 \pmod{m}$$
$$A - B \equiv 0 \pmod{m}$$

Note that $A + B \in \{0, \ldots, 2m-2\}$, therefore the only way for $A + B \equiv 0 (\bmod m)$ is for $A + B = \{0, m\}$. Similarly, $A - B \in \{-(m-1), \ldots, m-1\}$, therefore the only way for $A - B \equiv 0 (\bmod m)$ is for $A - B = 0$. Therefore, our probability simplifies to:

$$\mathbb{E}\left[\xi_{ij,k\ell}\xi_{kj,i\ell}\right] = \Pr[A = B = 0 \text{ or } A = B = \frac{m}{2}]$$
$$= \frac{1}{m^2} + \frac{\mathbb{1}[m \text{ is even}]}{m^2}.$$

The awkward indicator term occurs because if $m$ is odd, then $m/2$ is not an integer, therefore $A, B$ cannot take on that value and the expectation is $1/m^2$. If $m$ is even then they can and the expectation is $2/m^2$. Since $m$ is almost always even and a power of 2 for optimization purposes, we will assume the probability is $2/m^2$.

With the $\xi$ expectation handled, we move on to the sum over the tensors $\mathbf{X}, \mathbf{Y}$. Using the summation identity in Equation (18), the sum can be written as $T_1$ minus an extra off-diagonal like sum:

$$T_2 = \sum_{\substack{i \neq k \in [d] \\ j \neq \ell \in [d']}} \mathbf{X}_{ij}\mathbf{Y}_{k\ell}\mathbf{X}_{kj}\mathbf{Y}_{i\ell} \tag{21}$$

$$= \text{Equation (21)} + \sum_{\substack{i \in [d] \\ j \neq \ell \in [d']}} \mathbf{X}_{ij}^2\mathbf{Y}_{i\ell}^2 - \sum_{\substack{i \in [d] \\ j \neq \ell \in [d']}} \mathbf{X}_{ij}^2\mathbf{Y}_{i\ell}^2$$

$$= T_1 - \sum_{\substack{i \in [d] \\ j \neq \ell \in [d']}} \mathbf{X}_{ij}^2\mathbf{Y}_{i\ell}^2$$

$$= T_1 - \sum_{\substack{i \in [d] \\ j,\ell \in [d']}} \mathbf{X}_{ij}^2\mathbf{Y}_{i\ell}^2 + \sum_{\substack{i \in [d] \\ j \in [d']}} \mathbf{X}_{ij}^2\mathbf{Y}_{ij}^2$$

$$= T_1 - \text{tr}((\mathbf{X}\mathbf{X}^T) \circ (\mathbf{Y}\mathbf{Y}^T)) + \langle \mathbf{X} \circ \mathbf{X}, \mathbf{Y} \circ \mathbf{Y} \rangle$$

Note that the actual contribution to the sum is $\frac{2}{m^2}T_2$.

3: $(i = i' \neq k = k')$ and $(j = \ell \text{ and } \ell' = j')$

We begin by noting that the expectation over $\xi$ is the same for all elements in the sum:

$$\mathbb{E}\left[\xi_{ij,kj}\xi_{ij',kj'}\right] = \Pr[(H(i,j) = H(k,j) \text{ and } (H(i,j') = H(k,j')]$$

Expanding out the events in the probability, we have:

$$h_1(i) + h_2(j) \equiv h_1(k) + h_2(j) \pmod{m}$$
$$h_1(i) + h_2(j') \equiv h_1(k) + h_2(j') \pmod{m}.$$

Subtracting like terms, we have that the events simplify to:

$$h_1(i) - h_1(k) \equiv 0 \pmod{m}$$

The argument from here is identical to $T_1$ but with different indices: since $h_1$ is 2-wise independent and $i \neq k$, $h_1(i), h_1(k)$ can be treated as two independent uniform random variables in $[m]$, and therefore $h_1(i) - h_1(k) \bmod m$ is a uniform random variable in $\{0, 1, \dots, m-1\}$. Therefore the probability that it equals 0 is simply $1/m$, and the expectation over $\xi$ is:

$$\mathbb{E}\left[\xi_{ij,kj}\xi_{ij',kj'}\right] = \frac{1}{m}.$$

With the $\xi$ expectation handled, we move on to the sum over the tensors $\mathbf{X}, \mathbf{Y}$. The sum is a partner to $T_1$ since the arithmetic is identical but with different indices:

$$
\begin{aligned}
T_3 &= \sum_{\substack{i \neq k \in [d] \\ j,j' \in [d']}} \mathbf{X}_{ij}\mathbf{Y}_{kj}\mathbf{X}_{ij'}\mathbf{Y}_{kj'} \\
&= \sum_{i \neq k} \left(\sum_{j \in [d']} \mathbf{X}_{ij}\mathbf{Y}_{kj}\right)\left(\sum_{j' \in [d']} \mathbf{X}_{ij'}\mathbf{Y}_{kj'}\right) \\
&= \sum_{i \neq k} (\mathbf{X}\mathbf{Y}^T)_{ik}(\mathbf{X}\mathbf{Y}^T)_{ik} \\
&= \left\|\mathbf{X}\mathbf{Y}^T\right\|_F^2 - \sum_{i \in [d]} (\mathbf{X}\mathbf{Y}^T)_{ii}(\mathbf{X}\mathbf{Y}^T)_{ii} \\
&= \left\|\mathbf{X}\mathbf{Y}^T\right\|_F^2 - \|\mathrm{diag}(\mathbf{X}\mathbf{Y}^T)\|_2^2
\end{aligned}
$$

Note that the actual contribution to the sum is $\frac{1}{m}T_3$.

4: $(i = i' \neq k = k')$ and $(j = \ell' \neq \ell = j')$ We begin by noting that the expectation over $\xi$ is the same for all elements in the sum:

$$\mathbb{E}\left[\xi_{ij,k\ell}\xi_{i\ell,kj}\right] = \Pr[(H(i,j) = H(k,\ell) \text{ and } (H(i,\ell) = H(k,j)]$$

Expanding out the events in the probability, we have:

$$
\begin{aligned}
h_1(i) + h_2(j) &= h_1(k) + h_2(\ell) \quad \bmod m \\
h_1(i) + h_2(\ell) &= h_1(k) + h_2(j) \quad \bmod m.
\end{aligned}
$$

Moving terms around, we have that the events simplify to:

$$
\begin{aligned}
h_1(i) - h_1(k) &\equiv -(h_2(j) - h_2(\ell)) \quad \bmod m \\
h_1(i) - h_1(k) &\equiv h_2(j) - h_2(\ell) \quad \bmod m.
\end{aligned}
$$

The events are identical to those analysed in Case 2. Therefore we have:

$$\mathbb{E}\left[\xi_{ij,k\ell}\xi_{i\ell,kj}\right] = \frac{2}{m^2}$$

With the assumption that $m$ is even.

With the $\xi$ expectation handled, we move on to the sum over the tensors $\mathbf{X}, \mathbf{Y}$. This sum is a partner to $T_2$ in that just like how $T_2$ could be written as $T_1$ minus an off-diagonal like sum, so can $T_4$ be written as $T_3$ minus an off-diagonal

like sum using the summation identity in Equation (18):

$$T_4 = \sum_{\substack{i \neq k \in [d] \\ j \neq j' \in [d']}} \mathbf{X}_{ij} \mathbf{Y}_{kj'} \mathbf{X}_{ij'} \mathbf{Y}_{kj} \tag{22}$$

$$= Equation\ (22) + \sum_{\substack{i \neq k \in [d] \\ j \in [d']}} \mathbf{X}_{ij}^2 \mathbf{Y}_{kj}^2 - \sum_{\substack{i \neq k \in [d] \\ j \in [d']}} \mathbf{X}_{ij}^2 \mathbf{Y}_{kj}^2$$

$$= T_3 - \sum_{\substack{i \neq k \in [d] \\ j \in [d']}} \mathbf{X}_{ij}^2 \mathbf{Y}_{kj}^2$$

$$= T_3 - \sum_{\substack{i,k \in [d] \\ j \in [d']}} \mathbf{X}_{ij}^2 \mathbf{Y}_{kj}^2 + \sum_{\substack{i \in [d] \\ j \in [d']}} \mathbf{X}_{ij}^2 \mathbf{Y}_{ij}^2$$

$$= T_3 - \mathrm{tr}((\mathbf{X}^T \mathbf{X}) \circ (\mathbf{Y}^T \mathbf{Y})) + \langle \mathbf{X} \circ \mathbf{X}, \mathbf{Y} \circ \mathbf{Y} \rangle$$

Note that the actual contribution to the sum is $\frac{2}{m^2} T_4$.

5: $(i = k$ and $k' = i')$ and $(j = \ell' \neq \ell = j')$

We begin by noting that the expectation over $\xi$ is the same for all elements in the sum:

$$\mathbb{E}\left[\xi_{ij,i\ell}\xi_{i'\ell,i'j}\right] = \Pr[(H(i,j) = H(i,\ell) \text{ and } (H(i',\ell) = H(i',j)]$$

Expanding out the events in the probability, we have:

$$h_1(i) + h_2(j) \equiv h_1(i) + h_2(\ell) \pmod{m}$$
$$h_1(i') + h_2(\ell) \equiv h_1(i') + h_2(j) \pmod{m}.$$

Moving terms around, we have that the events simplify to:

$$h_2(j) - h_2(\ell) \equiv 0 \pmod{m}$$

This event is identical to the one in Case 1, and the expectation over $\xi$ is:

$$\mathbb{E}\left[\xi_{ij,i\ell}\xi_{i'\ell,i'j}\right] = \frac{1}{m}.$$

With the $\xi$ expectation handled, we move on to the sum over the tensors $\mathbf{X}, \mathbf{Y}$. This sum is another partner to $T_1$ in that the arithmetic is identical:

$$T_5 = \sum_{\substack{i,i' \in [d] \\ j \neq j' \in [d']}} \mathbf{X}_{ij} \mathbf{Y}_{ij'} \mathbf{X}_{i'j'} \mathbf{Y}_{i'j}$$

$$= \sum_{j \neq j' \in [d']} \left(\sum_{i \in [d]} \mathbf{X}_{ij} \mathbf{Y}_{ij'}\right) \left(\sum_{i' \in [d]} \mathbf{X}_{i'j'} \mathbf{Y}_{i'j}\right)$$

$$= \sum_{j \neq j' \in [d']} (\mathbf{X}^T \mathbf{Y})_{jj'} (\mathbf{X}^T \mathbf{Y})_{j'j}$$

$$= \sum_{j,j' \in [d']} (\mathbf{X}^T \mathbf{Y})_{jj'} (\mathbf{X}^T \mathbf{Y})_{j'j} - \sum_{j \in [d']} (\mathbf{X}^T \mathbf{Y})_{jj} (\mathbf{X}^T \mathbf{Y})_{jj}$$

$$= \mathrm{tr}((\mathbf{X}^T \mathbf{Y})^2) - \|\mathrm{diag}(\mathbf{X}^T \mathbf{Y})\|_2^2$$

Note that the actual contribution to the sum is $\frac{1}{m} T_5$. Note also that the diagonal term here is identical to the one in $T_1$.

6: (31) $(i = k' \neq k = i')$ and $(j = \ell$ and $\ell' = j')$

We begin by noting that the expectation over $\xi$ is the same for all elements in the sum:

$$\mathbb{E}\left[\xi_{ij,kj}\xi_{kj',ij'}\right] = \Pr[(H(i,j) = H(k,j) \text{ and } (H(k,j') = H(i,j')]$$

Expanding out the events in the probability, we have:

$$h_1(i) + h_2(j) = h_1(k) + h_2(j) \pmod{m}$$
$$h_1(k) + h_2(j') = h_1(i) + h_2(j') \pmod{m}.$$

Subtracting like terms, we have that the events simplify to:

$$h_1(i) - h_1(k) \equiv 0 \pmod{m}$$

This event is identical to the one in Case 3, and the expectation over $\xi$ is therefore:

$$\mathbb{E}\left[\xi_{ij,kj}\xi_{kj',ij'}\right] = \frac{1}{m}$$

With the $\xi$ expectation handled, we move on to the sum over the tensors $\mathbf{X}, \mathbf{Y}$. This sum is another partner to $T_1$ in that the arithmetic is identical:

$$
\begin{aligned}
T_6 &= \sum_{\substack{i \neq i' \in [d] \\ j,j' \in [d']}} \mathbf{X}_{ij}\mathbf{Y}_{i'j}\mathbf{X}_{i'j'}\mathbf{Y}_{ij'} \\
&= \sum_{i \neq i' \in [d]} \left(\sum_{j \in [d']} \mathbf{X}_{ij}\mathbf{Y}_{i'j}\right)\left(\sum_{j' \in [d']} \mathbf{X}_{i'j'}\mathbf{Y}_{ij'}\right) \\
&= \sum_{i \neq i' \in [d]} (\mathbf{X}\mathbf{Y}^T)_{ii'}(\mathbf{X}\mathbf{Y}^T)_{i'i} \\
&= \sum_{i,i \in [d]} (\mathbf{X}\mathbf{Y}^T)_{ii'}(\mathbf{X}\mathbf{Y}^T)_{i'i} - \sum_{i \in [d]} (\mathbf{X}\mathbf{Y}^T)_{ii}(\mathbf{X}\mathbf{Y}^T)_{ii} \\
&= \text{tr}((\mathbf{X}\mathbf{Y}^T)^2) - \|\text{diag}(\mathbf{X}\mathbf{Y}^T)\|_2^2 \\
&= \text{tr}((\mathbf{X}^T\mathbf{Y})^2) - \|\text{diag}(\mathbf{X}\mathbf{Y}^T)\|_2^2
\end{aligned}
$$

Where at the end, the cyclic property of the trace $(\text{tr}(PQ) = \text{tr}(QP))$ and the invariance of the trace under transposes $(\text{tr}(P) = \text{tr}(P^T))$ was used to turn the trace into the same one in $T_5$. Note that the actual contribution to the sum is $\frac{1}{m}T_6$. Note also that the diagonal term here is identical to the one in $T_3$.

7: $(i = i' \neq k = k')$ and $(j = j' \neq \ell = \ell')$

We begin by noting that the expectation over $\xi$ is the same for all elements in the sum:

$$\mathbb{E}\left[\xi_{ij,k\ell}\xi_{ij,k\ell}\right] = \Pr[(H(i,j) = H(k,\ell)]$$

Expanding out the events in the probability, we have:

$$h_1(i) + h_2(j) = h_1(k) + h_2(\ell) \pmod{m}$$

Moving terms around, we have that the event simplifies to:

$$h_1(i) - h_1(k) \equiv h_2(\ell) - h_2(j) \pmod{m}$$

Letting $A = h_1(i) - h_1(k) \mod m$ and $B = h_2(\ell) - h_2(j) \mod m$, we have that by the 2-wise independence of $h_1, h_2$, that $A, B$ are uniform random variables in $\{0, 1, \ldots, m-1\}$. Therefore, the probability that $A = B$ is simply $1/m$. And therefore:

$$\mathbb{E}\left[\xi_{ij,kj}\xi_{kj',ij'}\right] = \frac{1}{m}$$

With the $\xi$ expectation handled, we move on to the sum over the tensors $\mathbf{X}, \mathbf{Y}$. The arithmetic for simplifying this sum has no previous analogue. The summation identity Equation (17) was used to split the sum into 4 terms. Note that the two terms that appear with a minus sign are the same as the ones that appear in $T_2$ and $T_4$.

$$
\begin{aligned}
T_7 &= \sum_{\substack{i \neq k \in [d] \\ j \neq \ell \in [d']}} \mathbf{X}_{ij} \mathbf{Y}_{k\ell} \mathbf{X}_{ij} \mathbf{Y}_{k\ell} \\
&= \sum_{\substack{i,k \in [d] \\ j,\ell \in [d']}} (\mathbf{X}_{ij})^2 (\mathbf{Y}_{k\ell})^2 - \sum_{\substack{i \in [d] \\ j,\ell \in [d']}} (\mathbf{X}_{ij})^2 (\mathbf{Y}_{i\ell})^2 - \sum_{\substack{i,k \in [d] \\ j \in [d']}} (\mathbf{X}_{ij})^2 (\mathbf{Y}_{kj})^2 + \sum_{\substack{i \in [d] \\ j \in [d']}} (\mathbf{X}_{ij})^2 (\mathbf{Y}_{ij})^2 \\
&= \|\mathbf{X}\|_F^2 \|\mathbf{Y}\|_F^2 - \mathrm{tr}((\mathbf{X}\mathbf{X}^T) \circ (\mathbf{Y}\mathbf{Y}^T)) - \mathrm{tr}((\mathbf{X}^T\mathbf{X}) \circ (\mathbf{Y}^T\mathbf{Y})) + \langle \mathbf{X} \circ \mathbf{X}, \mathbf{Y} \circ \mathbf{Y} \rangle
\end{aligned}
$$

Note that the actual contribution to the sum is $\frac{1}{m} T_7$.

8: $(i = k' \neq k = i')$ and $(j = \ell' \neq \ell = j')$

We begin by noting that the expectation over $\xi$ is the same for all elements in the sum:

$$
\mathbb{E}\left[\xi_{ij,k\ell}\xi_{k\ell,ij}\right] = \Pr[H(i,j) = H(k,\ell)]
$$

Expanding out the events in the probability, we have:

$$
h_1(i) + h_2(j) \equiv h_1(k) + h_2(\ell) \pmod{m}
$$

This event is identical to the one in Case 7, and the expectation over $\xi$ is:

$$
\mathbb{E}\left[\xi_{ij,i\ell}\xi_{i'\ell,i'j}\right] = \frac{1}{m}.
$$

With the $\xi$ expectation handled, we move on to the sum over the tensors $\mathbf{X}, \mathbf{Y}$. The arithmetic in this sum is identical to $T_7$. Note that the two terms with a minus sign that appear here are the same as in $T_1$ and $T_3$. Note also that the final Hadamard term also appears in $T_2, T_4$, and $T_7$.

$$
\begin{aligned}
T_8 &= \sum_{\substack{i \neq k \in [d] \\ j \neq \ell \in [d']}} \mathbf{X}_{ij} \mathbf{Y}_{k\ell} \mathbf{X}_{k\ell} \mathbf{Y}_{ij} \\
&= \sum_{\substack{i,k \in [d] \\ j,\ell \in [d']}} \mathbf{X}_{ij} \mathbf{Y}_{ij} \mathbf{X}_{k\ell} \mathbf{Y}_{k\ell} - \sum_{\substack{i \in [d] \\ j,\ell \in [d']}} \mathbf{X}_{ij} \mathbf{Y}_{ij} \mathbf{X}_{i\ell} \mathbf{Y}_{i\ell} - \sum_{\substack{i,k \in [d] \\ j \in [d']}} \mathbf{X}_{ij} \mathbf{Y}_{ij} \mathbf{X}_{kj} \mathbf{Y}_{kj} + \sum_{\substack{i \in [d] \\ j \in [d']}} (\mathbf{X}_{ij})^2 (\mathbf{Y}_{ij})^2 \\
&= \langle \mathbf{X}, \mathbf{Y} \rangle^2 - \left\|\mathrm{diag}(\mathbf{X}\mathbf{Y}^T)\right\|_2^2 - \left\|\mathrm{diag}(\mathbf{X}^T\mathbf{Y})\right\|_2^2 + \langle \mathbf{X} \circ \mathbf{X}, \mathbf{Y} \circ \mathbf{Y} \rangle
\end{aligned}
$$

Note that the actual contribution to the sum is $\frac{1}{m} T_8$.

With all the cases handled, we now group the positive terms with a $2/m^2$ in front to get:

$$
P_1 = \left\|\mathbf{X}^T\mathbf{Y}\right\|_F^2 + \left\|\mathbf{X}\mathbf{Y}^T\right\|_F^2 + 2 \langle \mathbf{X} \circ \mathbf{X}, \mathbf{Y} \circ \mathbf{Y} \rangle
$$

and the negative terms to get:

$$
N_1 = \|\mathrm{diag}(\mathbf{X}^T\mathbf{Y})\|_2^2 + \|\mathrm{diag}(\mathbf{X}\mathbf{Y}^T)\|_2^2 + \mathrm{tr}((\mathbf{X}\mathbf{X}^T) \circ (\mathbf{Y}\mathbf{Y}^T)) + \mathrm{tr}((\mathbf{X}^T\mathbf{X}) \circ (\mathbf{Y}^T\mathbf{Y}))
$$

We do the same with the $1/m$ terms:

$$
\begin{aligned}
P_2 &= \left\|\mathbf{X}^T\mathbf{Y}\right\|_F^2 + \left\|\mathbf{X}\mathbf{Y}^T\right\|_F^2 + 2\mathrm{tr}((\mathbf{X}^T\mathbf{Y})^2) + \langle \mathbf{X}, \mathbf{Y} \rangle^2 + \|\mathbf{X}\|_F^2 \|\mathbf{Y}\|_F^2 + 2 \langle \mathbf{X} \circ \mathbf{X}, \mathbf{Y} \circ \mathbf{Y} \rangle \\
&= P_1 + 2\mathrm{tr}((\mathbf{X}^T\mathbf{Y})^2) + \langle \mathbf{X}, \mathbf{Y} \rangle^2 + \|\mathbf{X}\|_F^2 \|\mathbf{Y}\|_F^2
\end{aligned}
$$

and negative terms to get:

$$N_2 = 3\left[\|\text{diag}(\mathbf{X}^T\mathbf{Y})\|_2^2 + \|\text{diag}(\mathbf{X}\mathbf{Y}^T)\|_2^2\right] + \text{tr}((\mathbf{X}\mathbf{X}^T) \circ (\mathbf{Y}\mathbf{Y}^T)) + \text{tr}((\mathbf{X}^T\mathbf{X}) \circ (\mathbf{Y}^T\mathbf{Y}))$$
$$= N_1 + 2\left[\|\text{diag}(\mathbf{X}^T\mathbf{Y})\|_2^2 + \|\text{diag}(\mathbf{X}\mathbf{Y}^T)\|_2^2\right]$$

This means that:

$$\text{Var}[\langle \tilde{\mathbf{X}}, \tilde{\mathbf{Y}}\rangle] = \sum_{\substack{\mathbf{u}\neq\mathbf{v}\in[d]\times[d'] \\ \mathbf{u}'\neq\mathbf{v}'\in[d]\times[d']}} \mathbb{E}\left[S(\mathbf{u})S(\mathbf{v})S(\mathbf{u}')S(\mathbf{v}')\right] \mathbf{X_u}\mathbf{Y_v}\mathbf{X_{u'}}\mathbf{Y_{v'}}\mathbb{E}\left[\xi_{\mathbf{u},\mathbf{v}}\xi_{\mathbf{u}',\mathbf{v}'}\right]$$
$$= \frac{2}{m^2}(P_1 - N_1) + \frac{1}{m}(P_2 - N_2). \tag{23}$$

This concludes the first part of the proof, which was evaluating Equation (16) in closed form.

To upper-bound the variance, it is tempting to drop the negative terms and just upper bound $P_1, P_2$ with Cauchy-Schwartz, but a tighter bound can be achieved by noting that the term with a factor of 2 in $P_1$ is $\leq$ the trace terms in $N_1$. Using that $(\sum_j a_j)(\sum_\ell b_\ell) \geq \sum_j a_j b_j$:

$$\text{tr}((\mathbf{X}\mathbf{X}^T) \circ (\mathbf{Y}\mathbf{Y}^T)) = \sum_{i\in[d]}\left(\sum_{j\in[d']}\mathbf{X}_{ij}^2\right)\left(\sum_{\ell\in[d']}\mathbf{Y}_{i\ell}^2\right)$$
$$\geq \sum_{i\in[d],j\in[d']}\mathbf{X}_{ij}^2\mathbf{Y}_{ij}^2$$
$$= \langle \mathbf{X}\circ\mathbf{X}, \mathbf{Y}\circ\mathbf{Y}\rangle$$

The same argument yields that $\text{tr}((\mathbf{X}^T\mathbf{X}) \circ (\mathbf{Y}^T\mathbf{Y})) \geq \langle \mathbf{X}\circ\mathbf{X}, \mathbf{Y}\circ\mathbf{Y}\rangle$, therefore we have:

$$2\langle \mathbf{X}\circ\mathbf{X}, \mathbf{Y}\circ\mathbf{Y}\rangle \leq \text{tr}((\mathbf{X}\mathbf{X}^T) \circ (\mathbf{Y}\mathbf{Y}^T)) + \text{tr}((\mathbf{X}^T\mathbf{X}) \circ (\mathbf{Y}^T\mathbf{Y})).$$

Therefore, in order to upper bound the variance, the term $2\langle \mathbf{X}\circ\mathbf{X}, \mathbf{Y}\rangle$ can be dropped in $P_1$ and $P_2$. We can drop all the remaining terms in $N_1, N_2$ as well. We are left with:

$$\text{Var}[\langle \tilde{\mathbf{X}}, \tilde{\mathbf{Y}}\rangle] = \sum_{\substack{\mathbf{u}\neq\mathbf{v}\in[d]\times[d'] \\ \mathbf{u}'\neq\mathbf{v}'\in[d]\times[d']}} \mathbb{E}\left[S(\mathbf{u})S(\mathbf{v})S(\mathbf{u}')S(\mathbf{v}')\right] \mathbf{X_u}\mathbf{Y_v}\mathbf{X_{u'}}\mathbf{Y_{v'}}\mathbb{E}\left[\xi_{\mathbf{u},\mathbf{v}}\xi_{\mathbf{u}',\mathbf{v}'}\right]$$
$$= \frac{2}{m^2}(P_1 - N_1) + \frac{1}{m}(P_2 - N_2)$$
$$\leq \frac{2}{m^2}\left(\|\mathbf{X}^T\mathbf{Y}\|_F^2 + \|\mathbf{X}\mathbf{Y}^T\|_F^2\right)$$
$$+ \frac{1}{m}\left(\|\mathbf{X}^T\mathbf{Y}\|_F^2 + \|\mathbf{X}\mathbf{Y}^T\|_F^2 + 2\text{tr}((\mathbf{X}^T\mathbf{Y})^2) + \langle\mathbf{X},\mathbf{Y}\rangle^2 + \|\mathbf{X}\|_F^2\|\mathbf{Y}\|_F^2\right).$$

By the submultiplicative property of the Frobenius norm

$$\|\mathbf{X}^T\mathbf{Y}\|_F^2 \leq \|\mathbf{X}\|_F^2\|\mathbf{Y}\|_F^2, \quad \|\mathbf{X}\mathbf{Y}^T\|_F^2 \leq \|\mathbf{X}\|_F^2\|\mathbf{Y}\|_F^2.$$

and by Cauchy-Schwartz (and noting that $\text{tr}(A^2) = \langle A, A^\top\rangle$ then using the above bound):

$$\langle\mathbf{X},\mathbf{Y}\rangle^2 \leq \|\mathbf{X}\|_F^2\|\mathbf{Y}\|_F^2, \quad \text{tr}((\mathbf{X}^T\mathbf{Y})^2) \leq \|\mathbf{X}^T\mathbf{Y}\|_F^2 \leq \|\mathbf{X}\|_F^2\|\mathbf{Y}\|_F^2$$

We are left with:

$$\text{Var}[\langle \tilde{\mathbf{X}}, \tilde{\mathbf{Y}}\rangle] \leq \left(\frac{4}{m^2} + \frac{6}{m}\right)\|\mathbf{X}\|_F^2\|\mathbf{Y}\|_F^2. \tag{24}$$

We remark that this bound is strictly tighter than the Pham & Pagh (2025) bound of $8/m\|\mathbf{X}\|_F^2\|\mathbf{Y}\|_F^2$ for sketch dimension $m \geq 2$. As the sketch dimension $m$ goes to infinity, our bound is $33.\bar{3}\%$ tighter. At sketch dimensions above 400, the

improvement is already above $33\%$. Moreover, our bound is tight. The proof is relatively straightforward, in that we give an example for which the bound is saturated.

Let $\mathbf{x} = \mathbf{y} = \frac{1}{\sqrt{d}}\mathbf{1}_d$, where $\mathbf{1}_d$ is the $d$-dimensional vector of ones. Let $\mathbf{x}' = \mathbf{y}' = \frac{1}{\sqrt{d'}}\mathbf{1}_{d'}$. Now, define $\mathbf{X} = \mathbf{x} \otimes \mathbf{x}' = \frac{1}{\sqrt{dd'}}\mathbf{1}_{d\times d'}$, and $\mathbf{Y} = \mathbf{y} \otimes \mathbf{y}' = \frac{1}{\sqrt{dd'}}\mathbf{1}_{d\times d'}$, where $\otimes$ denotes the Kronecker product or outer product. This implies

$$\|\mathbf{X}^T\mathbf{Y}\|_F^2 = \|\mathbf{X}\mathbf{Y}^T\|_F^2 = \langle \mathbf{X}, \mathbf{Y}\rangle^2 = \mathrm{tr}((\mathbf{X}^T\mathbf{Y})^2) = \|\mathbf{X}\|_F^2\|\mathbf{Y}\|_F^2 = 1$$

$$\|\mathrm{diag}(\mathbf{X}^T\mathbf{Y})\|_2^2 = \mathrm{tr}((\mathbf{X}^T\mathbf{X}) \circ (\mathbf{Y}^T\mathbf{Y})) = \frac{1}{d'}, \quad \|\mathrm{diag}(\mathbf{X}\mathbf{Y}^T)\|_2^2 = \mathrm{tr}((\mathbf{X}\mathbf{X}^T) \circ (\mathbf{Y}\mathbf{Y}^T)) = \frac{1}{d}$$

$$\langle \mathbf{X} \circ \mathbf{X}, \mathbf{Y} \circ \mathbf{Y}\rangle = \frac{1}{dd'}$$

This means

$$P_1 = 2 + \frac{2}{dd'}, \quad N_1 = \frac{2}{d'} + \frac{2}{d}$$
$$P_2 = 6 + \frac{2}{dd'}, \quad N_2 = \frac{4}{d'} + \frac{4}{d}$$

and therefore, using Equation (23) we have that the variance equals:

$$\mathrm{Var}[\langle \tilde{\mathbf{X}}, \tilde{\mathbf{Y}}\rangle] = \frac{2}{m^2}(P_1 - N_1) + \frac{1}{m}(P_2 - N_2)$$
$$= \frac{4}{m^2}\left(1 + \frac{1}{dd'} - \frac{1}{d'} - \frac{1}{d}\right) + \frac{1}{m}\left(6 + \frac{2}{dd'} - \frac{4}{d'} - \frac{4}{d}\right)$$

Therefore, in the limit as $d, d' \to \infty$, we have that $\mathrm{Var}[\langle \tilde{\mathbf{X}}, \tilde{\mathbf{Y}}\rangle] = \frac{4}{m^2} + \frac{6}{m}$, which is equal to the bound in Equation (24) since $\|\mathbf{X}\|_F^2\|\mathbf{Y}\|_F^2 = 1$. $\qquad\square$

## C. Proof of Lemma 1

Here, we prove Lemma 1. This proof assumes familiarity with CountSketch and TensorSketch. See Appendix B for a self-contained overview of all required background.

**Lemma 1.** Fix examples $z, z'$, their SDI estimates $\mathcal{I}_t, \widetilde{\mathcal{I}}_t$ and checkpoints $\mathcal{W} = \{\mathbf{w}_k\}_{k=1}^K$. For each $k$, define the training-gradient vector $\mathbf{g}_k := \nabla_{\mathbf{w}_{\mathrm{body}}}\ell(\mathbf{w}_k; z)$, and for each $t \in \{1, \ldots, \tau\}$ define the test-step vector $\mathbf{p}_{k,t} := \phi_t(\mathbf{w}_k; z')$. Let $\mathcal{S}$ be the sketch map described above with sketch dimension parameter $m$, which we assume is even. Define the full and sketched dot products $\iota_{k,t} := \mathbf{g}_k \cdot \mathbf{p}_{k,t}$ and $\widetilde{\iota}_{k,t} := \widetilde{\mathbf{g}_k} \cdot \widetilde{\mathbf{p}_{k,t}}$. Then, $\mathbb{E}[\widetilde{\iota}_{k,t}] = \iota_{k,t}$ and $\mathrm{Var}(\widetilde{\iota}_{k,t}) \leq (4/m^2 + 6/m)\|\mathbf{g}_k\|_2^2\|\mathbf{p}_{k,t}\|_2^2$. Consequently, for each step $t$,

$$\mathbb{E}\left[\left(\widetilde{\mathcal{I}}_t - \mathcal{I}_t\right)^2\right] \leq \left(\frac{4}{m^2} + \frac{6}{m}\right)\left(\sum_{k=1}^K \eta_k\|\mathbf{g}_k\|_2\|\mathbf{p}_{k,t}\|_2\right)^2. \tag{12}$$

*Proof.* First, a remark on notation. In the main body we used $\mathbf{u} \cdot \mathbf{v}$ to denote the dot product, while in the proof below we use $\langle \mathbf{u}, \mathbf{v}\rangle$. This is done for notational clarity. Second, in Appendix B we use $\mathbf{X}_{ij}$ to denote the i-th row and j-th column of the matrix $\mathbf{X}$, and $\mathbf{x}_i$ to denote the i-th element of a vector $\mathbf{x}$. The proof below does not use this notation, instead $\mathbf{x}_i$ refers to the i-th vector in a sequence of vectors, and same for $\mathbf{X}_i$.

The proof for this lemma follows the following logic:

1. CountSketch is unbiased and has a controlled error. Formally, CountSketch with embedding dimension $m$, when applied to any two vectors $(\mathbf{x}, \mathbf{y})$ of the same shape, yields random embeddings $(\tilde{\mathbf{x}}, \tilde{\mathbf{y}})$ with the property that $\mathbb{E}[\langle \tilde{\mathbf{x}}, \tilde{\mathbf{y}}\rangle] = \langle \mathbf{x}, \mathbf{y}\rangle$ and $\mathrm{Var}[\langle \tilde{\mathbf{x}}, \tilde{\mathbf{y}}\rangle] \leq 2/m\|\mathbf{x}\|_2^2\|\mathbf{y}\|_2^2$.

2. TensorSketch is unbiased and has a controlled error. Formally, TensorSketch with embedding dimension $m$, when applied to any two matrices that are sums of outer products $(\mathbf{X}, \mathbf{Y})$ of the same shape, yields random embeddings $(\tilde{\mathbf{X}}, \tilde{\mathbf{Y}})$ with the property that $\mathbb{E}[\langle \tilde{\mathbf{X}}, \tilde{\mathbf{Y}}\rangle] = \langle \mathbf{X}, \mathbf{Y}\rangle$ and $\mathrm{Var}[\langle \tilde{\mathbf{X}}, \tilde{\mathbf{Y}}\rangle] \leq (4/m^2 + 6/m)\|\mathbf{X}\|_F^2\|\mathbf{Y}\|_F^2$.

3. The global sketch map $\mathcal{S}$, which flattens and concatenates sketched gradients across parameter tensors, preserves unbiasedness and bounded variance.

4. Summing over checkpoints preserves unbiasedness and bounded variance.

Item 1 and Item 2 follow from the results in Appendix B. Item 3 follows from linearity of expectation and variance when the sketched gradients are independent across parameter tensors. Concretely, suppose the training-gradient vector $\mathbf{g}_k$ is a concatenation of $\alpha$ vector-valued parameter gradients (which we denote by $\mathbf{x}$) and $\beta$ matrix-valued parameter gradients (which we denote by $\mathbf{X}$) and WLOG is of the form $\mathbf{g}_k^\top = (\mathbf{x}_1^\top, \mathbf{x}_2^\top, \ldots \mathbf{x}_\alpha, \mathbf{X}_1^\top, \mathbf{X}_2^\top \ldots \mathbf{X}_\beta^\top)$. By this, we mean that we concatenate all the vector-valued gradients first, and then flatten the matrix-valued parameter gradients and concatenate them to the end. Note here that $\mathbf{x}_1$ denotes the first vector in the concatenation $\mathbf{g}_k$, and not to the first element of the vector $\mathbf{x}$.

Since the sketch map $\mathcal{S}$ independently sketches parameter-level gradients to vectors of dimension $m$, let $\mathsf{CS}_1, \ldots, \mathsf{CS}_\alpha$ denote the $\alpha$ independent CountSketches (where by independent, we mean the underlying hashes are sampled independently), and $\mathsf{TS}_1, \ldots, \mathsf{TS}_\beta$ denote the $\beta$ independent TensorSketches. Formally, $\mathsf{TS}$ as defined in the main body acts on two vectors $\mathbf{u}, \mathbf{v}$ and sketches the outer product $\mathbf{u} \otimes \mathbf{v}$. However, below we abuse notation slightly and write $\mathsf{TS}(\mathbf{X})$ with the understanding that $\mathbf{X}$ is a sum of outer products (since it is a matrix-parameter gradient, see Proposition 1 for details) and $\mathsf{TS}$ is linear. The sketched training-gradient vector can be written as

$$\tilde{\mathbf{g}}_k = \mathcal{S}(\mathbf{g}_k) = (\mathsf{CS}_1(\mathbf{x}_1), \mathsf{CS}_2(\mathbf{x}_2)^\top, \ldots \mathsf{CS}_\alpha(\mathbf{x}_\alpha)^\top, \mathsf{TS}_1(\mathbf{X}_1)^\top, \mathsf{TS}_2(\mathbf{X}_2)^\top, \ldots \mathsf{TS}_\beta^\top(\mathbf{X}_\beta))$$
$$= (\tilde{\mathbf{x}}_1^\top, \tilde{\mathbf{x}}_2^\top, \ldots \tilde{\mathbf{x}}_\alpha^\top, \tilde{\mathbf{X}}_1^\top, \tilde{\mathbf{X}}_2^\top, \ldots \tilde{\mathbf{X}}_\beta^\top) \in \mathbb{R}^{(\alpha+\beta)m}.$$

We assume the same structure for the test-step vector $\mathbf{p}_{k,t}$:

$$\tilde{\mathbf{p}}_{k,t} = \mathcal{S}(\mathbf{p}_{k,t}) = (\mathsf{CS}_1(\mathbf{y}_1), \mathsf{CS}_2(\mathbf{y}_2)^\top, \ldots \mathsf{CS}_\alpha(\mathbf{y}_\alpha)^\top, \mathsf{TS}_1(\mathbf{Y}_1)^\top, \mathsf{TS}_2(\mathbf{Y}_2)^\top, \ldots \mathsf{TS}_\beta^\top(\mathbf{Y}_\beta))$$
$$= (\tilde{\mathbf{y}}_1^\top, \tilde{\mathbf{y}}_2^\top, \ldots \tilde{\mathbf{y}}_\alpha^\top, \tilde{\mathbf{Y}}_1^\top, \tilde{\mathbf{Y}}_2^\top, \ldots \tilde{\mathbf{Y}}_\beta^\top) \in \mathbb{R}^{(\alpha+\beta)m}$$

We now compute the full and sketched dot products $\iota_{k,t} := \langle \mathbf{g}_k, \mathbf{p}_{k,t} \rangle$ and $\widetilde{\iota}_{k,t} := \langle \widetilde{\mathbf{g}_k}, \widetilde{\mathbf{p}_{k,t}} \rangle$.

$$\iota_{k,t} = \langle \mathbf{g}_k, \mathbf{p}_{k,t} \rangle$$
$$\widetilde{\iota}_{k,t} = \langle \widetilde{\mathbf{g}_k}, \widetilde{\mathbf{p}_{k,t}} \rangle$$
$$= \sum_{i=1}^{\alpha} \langle \mathsf{CS}_i(\mathbf{x}_i), \mathsf{CS}_i(\mathbf{y}_i) \rangle + \sum_{j=1}^{\beta} \langle \mathsf{TS}_j(\mathbf{X}_j), \mathsf{TS}_j(\mathbf{Y}_j) \rangle$$

We observe that the inner product decomposes to the inner products of the respective sketches. Due to linearity of expectation, this means:

$$\mathbb{E}[\widetilde{\iota}_{k,t}] = \sum_{i=1}^{\alpha} \mathbb{E}[\langle \mathsf{CS}_i(\mathbf{x}_i), \mathsf{CS}_i(\mathbf{y}_i) \rangle] + \sum_{j=1}^{\beta} \mathbb{E}[\langle \mathsf{TS}_j(\mathbf{X}_j), \mathsf{TS}_j(\mathbf{Y}_j) \rangle]$$

Note that the randomness inside each expectation is over that respective CountSketch or TensorSketch. We can therefore use the fact that each CountSketch and TensorSketch is unbiased to conclude:

$$\mathbb{E}[\widetilde{\iota}_{k,t}] = \sum_{i=1}^{\alpha} \langle \mathbf{x}_i, \mathbf{y}_i \rangle + \sum_{j=1}^{\beta} \langle \mathbf{X}_j, \mathbf{Y}_j \rangle$$
$$= \langle \mathbf{g}_k, \mathbf{p}_{k,t} \rangle = \iota_{k,t}$$

Moreover, since each CountSketch/TensorSketch is independent, this means the variance of the sum is the sum of the variances:

$$\mathrm{Var}[\widetilde{\iota}_{k,t}] = \sum_{i=1}^{\alpha} \mathrm{Var}[\langle \mathsf{CS}_i(\mathbf{x}_i), \mathsf{CS}_i(\mathbf{y}_i) \rangle] + \sum_{j=1}^{\beta} \mathrm{Var}[\langle \mathsf{TS}_j(\mathbf{X}_j), \mathsf{TS}_j(\mathbf{Y}_j) \rangle]. \tag{25}$$

We can therefore use the variance bound for CountSketch from Pham & Pagh (2025) (restated in Lemma 2) and the novel variance bound for TensorSketch derived in Lemma 4[2] to upper bound the variance:

$$\mathrm{Var}[\widetilde{\iota}_{k,t}] \leq \frac{2}{m} \sum_{i=1}^{\alpha} \|\mathbf{x}_i\|_2^2 \|\mathbf{y}_i\|_2^2 + \left(\frac{4}{m^2} + \frac{6}{m}\right) \sum_{j=1}^{\beta} \|\mathbf{X}_j\|_F^2 \|\mathbf{Y}_j\|_F^2$$

$$= \sum_{i=1}^{\alpha} \frac{2}{m_i^{\mathsf{CS}}} \|\mathbf{x}_i\|_2^2 \|\mathbf{y}_i\|_2^2 + \sum_{j=1}^{\beta} \left(\frac{4}{m_j^2} + \frac{6}{m_j}\right) \|\mathbf{X}_j\|_F^2 \|\mathbf{Y}_j\|_F^2$$

$$\leq \left(\frac{4}{m^2} + \frac{6}{m}\right) \left(\sum_{i=1}^{\alpha} \|\mathbf{x}_i\|_2^2 \|\mathbf{y}_i\|_2^2 + \sum_{j=1}^{\beta} \|\mathbf{X}_i\|_F^2 \|\mathbf{Y}_j\|_F^2\right)$$

$$\leq \left(\frac{4}{m^2} + \frac{6}{m}\right) \|\mathbf{g}_k\|_2^2 \|\mathbf{p}_{k,t}\|_2^2$$

Where we used $2/m < 4/m^2 + 6/m$ and $\sum_i a_i b_i \leq (\sum_i a_i)(\sum_j b_j)$ to upper bound the variance. Note that this bound in tight in that we have equality when there are no vector-valued parameters ($\alpha = 0$), there is one matrix-valued parameter ($\beta = 1$), and $\mathbf{g}_k, \mathbf{p}_{k,t}$ are outer products of parallel unit-RMS norm vectors as described in Lemma 4.

Now for Item 4. We have that:

$$\mathcal{I}_t(z, z') = \sum_{k=1}^{K} \eta_k \iota_{k,t}$$

$$\widetilde{\mathcal{I}}_t(z, z') = \sum_{k=1}^{K} \eta_k \widetilde{\iota}_{k,t}$$

By linearity of expectation and unbiasedness, $\mathbb{E}[\widetilde{\mathcal{I}}_t(z, z')] = \mathcal{I}_t(z, z')$. Therefore:

$$\mathbb{E}[\left(\widetilde{\mathcal{I}}_t(z, z') - \mathcal{I}_t(z, z')\right)^2] = \mathrm{Var}(\widetilde{\mathcal{I}}_t(z, z'))$$

Note that we use the same sketches for every checkpoint $k$, which means the sketched gradients across checkpoints are correlated. In contrast to Equation (25), here the variance of the sum is not the sum of the variances. Instead, we include the covariance terms, and use Cauchy-Schwarz to upper bound the covariances:

$$\mathbb{E}[\left(\widetilde{\mathcal{I}}_t(z, z') - \mathcal{I}_t(z, z')\right)^2] = \mathrm{Var}(\widetilde{\mathcal{I}}_t(z, z'))$$

$$= \sum_{k=1}^{K} \eta_k^2 \mathrm{Var}(\widetilde{\iota}_{k,t}) + \sum_{i \neq j \in [K]} \eta_i \eta_j \, \mathrm{Cov}(\widetilde{\iota}_{i,t}, \ \widetilde{\iota}_{j,t})$$

$$\leq \sum_{k=1}^{K} \eta_k^2 \mathrm{Var}(\widetilde{\iota}_{k,t}) + \sum_{i \neq j \in [K]} \eta_i \eta_j \sqrt{\mathrm{Var}(\widetilde{\iota}_{i,t}) \mathrm{Var}(\widetilde{\iota}_{j,t})}$$

$$\leq \left(\frac{4}{m^2} + \frac{6}{m}\right) \sum_{k=1}^{K} \eta_k^2 \|\mathbf{g}_k\|_2^2 \|\mathbf{p}_{k,t}\|_2^2$$

$$+ \left(\frac{4}{m^2} + \frac{6}{m}\right) \sum_{i \neq j \in [K]} \eta_i \eta_j \|\mathbf{g}_i\|_2 \|\mathbf{p}_{i,t}\|_2 \|\mathbf{g}_j\|_2 \|\mathbf{p}_{j,t}\|_2$$

$$= \left(\frac{4}{m^2} + \frac{6}{m}\right) \left(\sum_{k=1}^{K} \eta_k \|\mathbf{g}_k\|_2 \|\mathbf{p}_{k,t}\|_2\right)^2$$

$\square$

---

[2]The bound in Lemma 4 does not assume $\mathbf{X}, \mathbf{Y}$ are sums of outer products, and applies more generally to TensorSketches of any two matrices.

# D. Further experimental details

## D.1. Experimental details: scalability and correctness

We validate the scalability, fidelity, and systems overhead of SDI. All runs use PyTorch in full float32 precision and full backpropagation through the unrolled recurrence (no truncation).

**Model** We instantiate a looped GPT-style model (from the nanochat GPT implementation (Karpathy, 2025)) with sequence length $T=128$ and loop horizon $\tau=32$ recurrent iterations. The architecture follows a *prelude–recurrent core–coda* pattern with 2 prelude blocks, a recurrent core consisting of a stack of 4 transformer blocks applied $\tau$ times, and 2 coda blocks, giving an effective depth of $2 + 4\tau + 2 = 132$ blocks. We set `bptt_k=None` to enforce full BPTT through all $\tau$ iterations. Unless otherwise stated, SDI targets the loop-body parameters corresponding to the recurrent core plus the injection adapter.

**Data and loss** Because this benchmark evaluates sketch fidelity and systems overhead (not generalisation), we use synthetic random token sequences. For each trial we sample `idx` and `targets` uniformly at random from the model vocabulary to form a batch of shape $(B, T)$. We compute a per-example loss vector by requesting per-token cross-entropy losses (no reduction) and averaging over tokens, yielding $\ell_i \in \mathbb{R}$ for each example $i \in \{1, \ldots, B\}$.

**Baselines and estimators** We compare: (i) Full Gradient TracIn/SDI, which materializes full per-example stepwise gradients, and (ii) Projected TracIn/SDI, which computes stepwise features *during* backprop using TensorSketch/CountSketch without instantiating full per-example gradients as described above. For the projected estimator, we sweep sketch dimension $m \in \{256, 512, 1024, 2048, 4096, 8192\}$ (default), and average results over 10 independent sketch seeds. For a fixed batch, we treat the batch as both the "train" and "query" sets and compute SDI producing an SDI tensor in $\mathbb{R}^{B \times B \times \tau}$; the corresponding TracIn matrix is obtained by summing over steps.

**Fidelity metrics** For SDI and TracIn, we report relative Frobenius error: $\|\widehat{\text{SDI}} - \text{SDI}\|_F / \|\text{SDI}\|_F$ and $\|\widehat{\text{TracIn}} - \text{TracIn}\|_F / \|\text{TracIn}\|_F$, where the unprojected full-gradient result is treated as ground truth. We also fit a log–log slope of mean SDI error versus $m$ to verify the expected $\mathcal{O}(m^{-1/2})$ scaling, the result of which is shown in Figure 5. As expected, the relative scales as $\mathcal{O}(1/\sqrt{m})$.

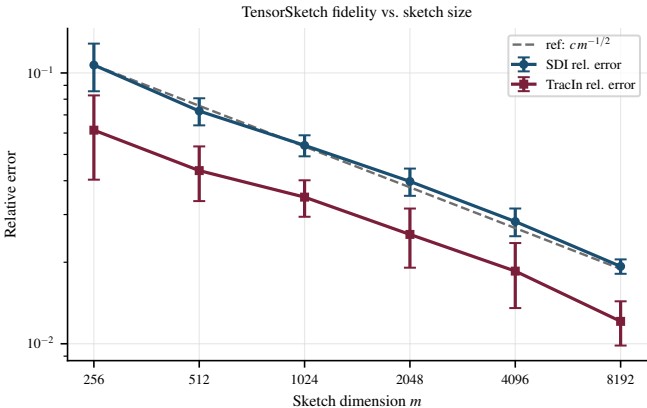

*Figure 5.* Empirical error scaling w.r.t sketch dimension $m$. Average and standard deviation over 10 trials.

**Conservation check** To empirically verify sketched conservation, we select representative loop-body matrix parameters and compare: (1) a *direct* TensorSketch applied to the summed full matrix gradient $\sum_{t=1}^{\tau} \phi_t$, versus (2) the sum of the per-step sketched gradients $\sum_{t=1}^{\tau} \widetilde{\phi}_t$ produced by `ProjectedTracInSDI`. We report the maximum absolute discrepancy and relative $\ell_2$ discrepancy.

**Memory and runtime** On CUDA (NVidia RTX A6000), we quantify the maximum feasible batch size under each method via an out-of-memory search (batch sizes starting at 4, increasing in steps of 4). We additionally measure (i) the overhead of projected featurization relative to a plain backward pass on the same batch, and (ii) the added wall-clock time per checkpoint relative to an inference-only forward pass, where per-checkpoint SDI time is defined as one projected featurization plus SDI/TracIn reductions on that batch.

## D.2. Additional efficiency and scaling analysis

SDI is a post-hoc analysis method applied to saved checkpoints, so it does not change the cost of the original training run. The relevant comparison is therefore between a standard forward/backward pass used for gradient featurisation and the additional sketching and reduction work required by SDI. Table 3 gives an operator-level breakdown for the projected SDI pass. The dominant additional cost is TensorSketch construction, while the final SDI inner products are negligible.

*Table 3.* **Timing breakdown of projected SDI featurisation.** The first block reports the share of total featurisation time. The second block decomposes TensorSketch time only.

| Timing level | Component | Share |
|---|---|---|
| Full SDI pass | Standard forward+backward | 51.2% |
| Full SDI pass | TensorSketch construction | 48.4% |
| Full SDI pass | SDI inner products/reductions | 0.4% |
| TensorSketch only | Scatter-add | 51.8% |
| TensorSketch only | FFT | 18.9% |
| TensorSketch only | Pointwise multiply + iFFT | 29.2% |

We also measured scaling with model size at fixed $\tau=32$, sketch dimension $m=2048$, sequence length $L=128$, FP32 precision, and a single A100-80GB GPU. As shown in Table 4, the relative sketch cost decreases as model width grows. This matches the computational scaling: for fixed $m$, TensorSketch updates for transformer linear layers scale roughly as $\mathcal{O}(L(d + m \log m))$, whereas the standard backward pass scales roughly as $\mathcal{O}(Ld^2)$ in the width-dominated regime.

*Table 4.* **Scaling with model size.** All runs use $\tau=32$, $m=2048$, $L=128$, FP32 precision, and a single A100-80GB GPU. Batch size $B$ is reduced as needed to fit memory.

| Model | $B$ | Fwd+bwd (ms) | Sketch (ms) | Rel. sketch cost |
|---|---|---|---|---|
| 77M | 16 | 454 | 526 | +116% |
| 135M | 8 | 474 | 447 | +94% |
| 386M | 4 | 840 | 313 | +37% |
| 617M | 4 | 1365 | 354 | +26% |
| 1.2B | 2 | 1592 | 291 | +18% |

Finally, we measured scaling with the number of recurrent loop steps $\tau$ for the 135M model. Both the standard forward/backward pass and TensorSketch construction scale approximately linearly in $\tau$. The relative sketch overhead increases at large $\tau$ because the forward/backward baseline includes fixed non-looped costs such as embeddings, prelude blocks and coda blocks, whereas the sketching work is concentrated in the recurrent body.

*Table 5.* **Scaling with loop count $\tau$.** All runs use the 135M model, $m=2048$, $L=128$, FP32 precision, and a single A100-80GB GPU.

| $\tau$ | Effective depth | $B$ | Fwd+bwd (ms) | Sketch (ms) |
|---|---|---|---|---|
| 4 | 20 | 8 | 84 | 63 |
| 32 | 132 | 8 | 474 | 447 |
| 64 | 260 | 4 | 564 | 961 |

## D.3. Experimental details: SDI as a mechanistic interpretability hypothesis generator (parity)

**Task and sequence format** Parity examples are token sequences over a vocabulary of size 6. Inputs are bit strings (tokens 0/1) of variable length $n$, followed by an "=" token (token 2) and padding (token 3) to a fixed sequence length $T=n+2$. Labels place the parity bit (0/1) at the "=" position; non-answer positions are masked out of the loss via a per-token mask. The model receives an additional per-example integer `batch_nums` equal to the number of digits $n$; in our implementation `batch_nums` is *1-indexed* and indicates the loop iteration at which the model state is read out. The methodology here is identical to Fan et al. (2024).

**Model and recurrence** We use a `LoopedTransformer` as described in the parity task experiments of Fan et al. (2024) with a linear read-in map (`linear_embedding=True`), embedding width $d_{\text{model}}{=}256$, one decoder block (`n_layer=1`) and 64 attention heads. The model executes exactly $T$ recurrent iterations, with additive input injection at every iteration. The model captures (and later decodes) the state at iteration `batch_nums` and ignores later iterations. Thus, for an $n$-bit parity instance, the model loops for $\tau = T = n + 2$ iterations but uses the hidden state from iteration $n$ for prediction; stepwise influence beyond the readout step is therefore zero.

**Subsampled train/test sets for influence** To make SDI computation lightweight while preserving coverage over difficulty (sequence length), we subsample: (i) 512 training examples, constructed as 128 from each tercile of the training length distribution plus an additional 128 examples at length exactly 20, and (ii) 384 test examples sampled from the held-out test set (all length 40 in this setup). We also create two *special* alternating probes $(0101\ldots)$ and $(1010\ldots)$ of length 40 (with $T{=}42$ total tokens) for the probe experiment in the main body.

**SDI computation** We compute SDI/TracIn on the loop-body parameters using Projected TracIn/SDI with sketch dimension $m{=}2048$ and per-example masked cross-entropy loss. We aggregate across all checkpoints; checkpoint weights $\eta_k$ are taken from the optimizer nominal learning rate. We compute: (i) train$\rightarrow$test SDI/TracIn, (ii) train$\rightarrow$train SDI/TracIn for self-influence (diagonal), and (iii) train$\rightarrow$special SDI trajectories for the alternating probes.

**Mechanistic readouts** For a fixed alternating probe, we form a *summed SDI profile* by aggregating across training points: $\text{SDI}_\Sigma(t) := \sum_{z \in \mathcal{D}_{\text{train}}} \text{SDI}(z, \text{probe})_t$. In parallel, we explicitly unroll the recurrent computation for all $T{=}42$ iterations and record: (i) the answer-position hidden state $h_t$ after each loop iteration, and (ii) the logit margin between the true parity class and the incorrect class at each iteration. We quantify periodic structure by (a) autocorrelation of the margin curve and (b) cosine similarity $\cos(h_t, h_{t+4})$. We then compute a 2D PCA embedding of $\{h_t\}_{t=1}^T$ and cluster the trajectory with $k$-means ($k{=}4$) to obtain a discrete state sequence and an empirical transition matrix (row-normalized).

**Proxy circuit evaluation** To turn the discrete-state hypothesis into an interpretable proxy, we learn a cluster$\rightarrow$parity lookup table from a small calibration set of alternating sequences (256 examples with random length in $[2, 40]$). We evaluate this proxy on a larger alternating evaluation set (1024 examples), including an in-distribution split (length $\leq 20$) and an out-of-distribution split (length $> 20$). To diagnose when the proxy applies beyond the alternating family, we also report a *selective* proxy accuracy by thresholding the $k$-means distance to the nearest centroid, using the 95th percentile distance on the alternating calibration set as the "on-manifold" threshold.

### D.4. Experimental details: scaling laws of loop compute (Sudoku)

**Dataset and difficulty** We use the SATNet Sudoku tensors from Wang et al. (2019), each of shape $(10{,}000, 9, 9, 9)$. We define puzzle difficulty as the number of missing cells ("blanks"), computed as the number of grid positions whose 9-way input vector is all zeros.

**Train/test split (non-IID by difficulty)** We sort puzzles by missing-cell count and assign the easiest $80\%$ to train and the hardest $20\%$ to test, then shuffle each split with the same seed. All loop-scaling evaluations in this section are performed on the *full hard test split*.

**Model and training-time loop sampling** We instantiate a looped transformer taking inspiration from Yang et al. (2023b) using non-causal/bidirectional attention with learned positional embeddings with $d_{\text{model}}{=}128$, 1 decoder block, 4 attention heads, and a recurrent loop horizon with training mean 32. Training samples the number of loops per batch from a Poisson log-normal distribution (clipped to $1 \leq r \leq 64$) as described in McLeish et al. (2025); Geiping et al. (2025).

**Board accuracy and evaluation across loop budgets** Given model logits of shape $(B, 81, 9)$, we predict a board by $\arg\max$ over the last dimension. A board is counted correct iff *all originally blank cells* are predicted correctly:

$$\texttt{board\_correct} = \big((\hat{y} = y) \lor \neg\texttt{blank\_mask}\big) \text{ all over cells.}$$

We evaluate board accuracy at multiple *test-time loop budgets* by re-running the same checkpoint with an overridden loop count.

**SDI energy curves** To relate compute scaling to data influence, we compute SDI on a *fixed* analysis horizon (default $\tau{=}32$) and summarize stepwise magnitude as an "energy" curve. We define per-test-example SDI energy

$$e_j(t) = \sum_{i \in \text{sampled train}} \big|\text{SDI}(z_i, z_j')_t\big|.$$

We then aggregate $e_j(t)$ across test examples within each difficulty bin and plot the median and IQR versus $t$ on a log-scaled $y$-axis. (Panel A uses the full hard test split for accuracy; Panel B uses the SDI computation subset described below.)

**Difficulty bins** For visualization, we bin test puzzles by missing-cell counts. The plotting script uses an "auto" binning heuristic that produces three approximately equally sized bins for the common hard-split support $\{46,47,48,49,50\}$: (46–47), (48), and (49–50).

### D.5. Experimental details: step-resolved influence across Sudoku difficulty

**SDI artifacts and sampled subsets** We recreate the same non-IID easy/hard split as in training (easy 80% train, hard 20% test by missing cells), then subsample fixed-size subsets from each split: 512 training puzzles and 512 test puzzles. Sampling is stratified by missing-cell count, allocating samples as uniformly as possible across the missing-count strata subject to each stratum's capacity. We compute influence on a fixed analysis horizon $\tau{=}32$ using Projected TracIn/SDI with sketch dimension $m{=}2048$ targeting the recurrent block parameters.

**Loss and evaluation unit** For each Sudoku puzzle, the per-example loss is the mean cross-entropy over *blank* cells only (cells that are missing in the input), matching the standard "solve the blanks" evaluation. All influence quantities treat an example as a full Sudoku board (81-way token sequence).

**Self-influence and cross-influence** From the train→train TracIn matrix, we define self-influence for each training puzzle $z$ as the diagonal entry $\mathrm{TracIn}(z, z)$. From the train→test TracIn matrix, we define the cross-influence mass of a training puzzle

$$C(z) \;=\; \sum_{z' \in \mathcal{D}_{\text{test}}} \big|\mathrm{TracIn}(z, z')\big|.$$

We report correlations of these quantities with training difficulty (missing-cell count) and with each other.

**Train/test difficulty groups** We define *easy train* as missing $\leq 42$ and *hard train* as missing $\geq 45$ (leaving 43–44 as a middle band), and we evaluate on two test difficulty bins: (46–47 missing) as "easy test" and (49–50 missing) as "hard test". These groupings are applied within the sampled train/test subsets.

**Share of $|\mathrm{TracIn}|$ mass by training group.** For a fixed test puzzle $z'$, let $M(z') = \sum_z |\mathrm{TracIn}(z, z')|$ be its total absolute influence mass over the sampled training set. For a training group $G$, define its share on $z'$ as

$$\mathrm{Share}_G(z') \;=\; \frac{\sum_{z \in G} |\mathrm{TracIn}(z, z')|}{M(z')}.$$

We report the mean of $\mathrm{Share}_G(z')$ over test puzzles within each test difficulty bin.

**Step-resolved SDI timing (SDI energy and late mass)** Given the SDI tensor $\mathrm{SDI}(z, z') \in \mathbb{R}^\tau$, we define per-test SDI energy curves by summing absolute SDI across training points:

$$e_{G,z'}(t) \;=\; \sum_{z \in G} \big|\mathrm{SDI}(z, z')_t\big|.$$

To summarise *when* influence acts, we normalise per test example to a distribution $p_{G,z'}(t) = e_{G,z'}(t)/\sum_{s=1}^{\tau} e_{G,z'}(s)$ and compute timing statistics. In particular, we define "late" steps as $t \geq 17$ (i.e., indices $\geq 16$ in 0-based indexing) and report the *SDI late mass*

$$\mathrm{LateMass}_G(z') \;=\; \sum_{t=17}^{\tau} p_{G,z'}(t),$$

averaged over test puzzles in each difficulty bin. We also compute auxiliary timing summaries (e.g., centre of mass and CDF crossing steps) and validate the conservation identity $\mathrm{TracIn}(z, z') = \sum_{t=1}^{\tau} \mathrm{SDI}(z, z')_t$ numerically (maximum absolute discrepancy).

**Statistical tests** We use Spearman rank correlation for monotone association claims, and Welch two-sample $t$-tests (and paired $t$-tests where appropriate for per-test paired shares) to report $p$-values for group comparisons. Statistical significance is set at $p < 0.05$.

### D.6. Robustness of SDI energy aggregations

The main text uses the default SDI energy aggregation

$$e_j(t) = \sum_i |\text{SDI}(z_i, z'_j)_t|.$$

We tested whether the conclusions depend on this aggregation by comparing it to RMS energy, a top-$k$ mean with $k=50$, and a positive-only sum. We also compared to a per-step gradient-norm baseline $\|\phi_t(z')\|_2$, which measures step-local gradient magnitude without reference to training data. Table 6 reports Spearman rank correlations of per-example rankings with the default aggregation. All SDI-based aggregations are highly concordant and preserve the hard>easy late-step ordering, whereas the gradient-norm baseline is nearly uncorrelated and reverses this ordering.

Table 6. **Robustness of SDI energy aggregation.** Rank correlations are with the default aggregation $\sum_i |\text{SDI}|$. The gradient-norm baseline is included to test whether step-local gradient magnitude alone explains the SDI energy curves.

| Method | Definition | Spearman $\rho$ |
|---|---|---|
| Default $L_1$ energy | $\sum_i |\text{SDI}(z_i, z')_t|$ | 1.00 |
| RMS | $\left(\frac{1}{n} \sum_i \text{SDI}(z_i, z')_t^2\right)^{1/2}$ | 0.96 |
| Top-50 mean | mean of top 50 values of $|\text{SDI}(z_i, z')_t|$ | 0.92 |
| Positive-only sum | $\sum_i \max(\text{SDI}(z_i, z')_t, 0)$ | 0.96 |
| Gradient norm | $\|\phi_t(z')\|_2$ | 0.07 |

### D.7. Counterfactual removal protocol for Sudoku

For the counterfactual removal experiment in Table 2, we first compute late-step SDI mass on a held-out validation split and rank training puzzles by their validation influence. We then remove the top $30\%$ of training puzzles by late-step SDI mass, retrain the Sudoku model from scratch, select checkpoints using validation accuracy, and report final results on a disjoint hard-puzzle test split. All numbers are averaged over $8$ independent runs. A top-decile intervention did not produce a stable effect, likely because Sudoku has substantial shared structure across puzzles; the stronger $30\%$ removal exposes the functional role of late-step influential examples while preserving enough data for successful retraining. For comparison, removing early-step influential examples produced broader degradation and non-convergence in $2/8$ runs, consistent with early-step examples supporting more foundational aspects of the computation.

### D.8. Signal cancellation diagnostics

To quantify signal cancellation, define the cancellation ratio

$$R(z, z') = \frac{\sum_{t=1}^{\tau} |\mathcal{I}_t(z, z')|}{|\sum_{t=1}^{\tau} \mathcal{I}_t(z, z')|}.$$

The ratio equals 1 when all step contributions share the same sign and grows when positive and negative stepwise effects cancel in scalar TracIn. On Sudoku, among pairs with substantial step-level influence (top quartile by $\sum_t |\mathcal{I}_t|$), the mean cancellation ratio is $\bar{R}=5.38$, compared to an overall dataset average of 4.3. Moreover, $4.1\%$ of pairs satisfy $R>5$, meaning that more than $80\%$ of the step-level influence magnitude would be lost to cancellation in scalar TracIn. When restricting to pairs where both early-step contributions ($t\leq16$) and late-step contributions ($t\geq17$) are individually large, the cancellation rate grows with test difficulty: it is $5.0\%$ for the easiest test puzzles and $9.7\%$ for hard test puzzles. This suggests that hard Sudoku puzzles create more opportunities for a training puzzle to help early constraint propagation while hindering late iterative refinement.

### D.9. AdamW-preconditioned SDI sanity check

Modern LLM training often uses AdamW rather than SGD, so we tested whether an AdamW-style preconditioned inner product changes the examples identified by TracIn/SDI. Let $\mathbf{v}_k$ denote the AdamW second-moment accumulator at

checkpoint $k$. We define the preconditioned step score

$$\mathcal{I}_t^{\mathrm{Adam}}(z, z') = \sum_{k=1}^{K} \eta_k \left\langle \frac{\nabla_{\mathbf{w}_{\mathrm{body}}} \ell(\mathbf{w}_k; z)}{\sqrt{\mathbf{v}_k + \epsilon}}, \frac{\phi_t(\mathbf{w}_k; z')}{\sqrt{\mathbf{v}_k + \epsilon}} \right\rangle,$$

where division is elementwise. Because the preconditioner is fixed at each checkpoint and applied linearly to each gradient vector, the conservation identity across steps is preserved. On Sudoku, where the effective per-coordinate learning rate varies by up to $131\times$, the standard and preconditioned rankings remain very close: Spearman $\rho{=}0.979$ for scalar TracIn, Spearman $\rho{=}0.934$ for SDI across steps, and $93.7\%$ overlap among the top-5% most influential examples. Thus, preconditioning refines absolute magnitudes but does not materially change the ranking-based analyses reported in this work.

### D.10. Experimental details: Recursive Nanochat

**Model and recurrence** We analyze a 328.3M-parameter recursive GPT-style chat model (Karpathy, 2025) with a *prelude–recurrent core–coda* decomposition and an input injection adapter. The model has 2 prelude blocks, a recurrent core consisting of a stack of 4 transformer blocks reused across $r$ recurrent iterations, and 2 coda blocks; for fixed $r$, the effective depth is $2 + 4r + 2$. Training uses truncated BPTT with $k{=}4$, implemented by detaching the recurrent state for iterations earlier than the last $k$.

**Training stages and TracIn checkpoints** We follow the standard nanochat pipeline (pretraining, mid-training and supervised fine-tuning). During the GSM8K-containing stages (mid-training and supervised fine-tuning), we periodically save TracIn checkpoints along with optimizer learning-rate metadata and aggregate influence across all available checkpoints in each stage (82/71 checkpoints, respectively). Checkpoint weights $\eta_k$ are taken from the recorded learning rates.

**Train sources and query set** For mid training and supervised fine-tuning, we study cross-influence from GSM8K training into GSM8K test loss. We construct fixed tokenized sample sets by uniformly sampling 1024 conversations, truncating each rendered conversation to at most 2048 tokens. The mid training sources are SmølTalk, MMLU, GSM8K, Identity, Simple Spelling, and SpellingBee. The supervised fine-tuning training sources are ARC-Easy, ARC-Challenge, GSM8K, SmølTalk, Identity, Simple Spelling, and SpellingBee.

**Losses** All losses are per-example means of the token-level cross-entropy over *valid* (unmasked) targets. For MID training examples, we use next-token loss over all tokens in the conversation. For SFT training examples, we use an assistant-only mask (matching the SFT training objective).

**SDI computation** We compute Projected TracIn/SDI targeting the recurrent core and injection adapter parameters, with sketch dimension $m{=}2048$, evaluating fixed recurrence counts $r \in \{2, \dots, 16\}$. By default, SDI uses the checkpoint's truncated-BPTT setting; when $r > k$, only the last $k$ loop steps contribute nonzero SDI and earlier steps are zero by construction.

### D.11. Nanochat robustness and loop-step probing

We repeated the Nanochat SDI analysis with two additional random seeds. Table 7 reports the last-step and last-four-step SDI shares across three seeds. The late-step concentration is stable across seeds and across analysis horizons. The approximately geometric late tail is also robust, with late-tail log-linear $R^2 \approx 0.996$ in mid-training and $R^2 \approx 0.95$ in supervised fine-tuning.

*Table 7.* **Nanochat late-step SDI robustness across seeds.** Values are mean±standard deviation over three seeds.

| Stage | Analysis horizon $\tau$ | Last-step share | Last-four-step share |
|---|---|---|---|
| Mid-training | 8 | $0.532 \pm 0.010$ | $0.885 \pm 0.006$ |
| Mid-training | 16 | $0.506 \pm 0.017$ | $0.824 \pm 0.013$ |
| SFT | 8 | $0.604 \pm 0.026$ | $0.989 \pm 0.002$ |
| SFT | 16 | $0.606 \pm 0.028$ | $0.987 \pm 0.001$ |

To test whether the model represents loop progress in the latent state, we trained a linear probe to decode the loop step within the TBPTT window from activations at a fixed token position. The probe achieves $98.56\%$ accuracy on a held-out split, compared to $25.00\%$ chance accuracy for a randomly initialised model with identical architecture. Because the token

position is fixed, this result is not explained by standard token positional embeddings alone and supports the hypothesis that the trained recurrent model internally represents loop-step progress.

