# OpenReview forum: "Step-Resolved Data Attribution for Looped Transformers"
_ICML.cc/2026/Conference — ICML 2026 regular_

### Official Review · Reviewer_rstq · 2026-03-11

**Soundness:** 3
**Presentation:** 2
**Significance:** 3
**Originality:** 3
**Overall Recommendation:** 4
**Confidence:** 4

**Summary:**

This paper studies training-data attribution for looped transformers, where a shared transformer block is applied recurrently for multiple iterations in a looping fashion. The author propose Step-Decomposed Influence (SDI), which decomposes TracIn-style influence scores into a per-step trajectory over the recurrent computation, with a conservation identity showing that summing the stepwise influences recovers the body-parameter TracIn score. To make this practical at transformer scale, a sketch-during-backprop based on TensorSketch is proposed so that per-example gradients need not be cached, thus reducing memory footprint. Experiments on the parity tasks and Nanochat, to some extend, demonstrate several interpretability case studies and scalability.

**Compliance With Llm Reviewing Policy:**

Affirmed.

**Final Justification:**

My main concerns are now adequately addressed. In particular, the new counterfactual retraining results substantially strengthen the causal interpretation of SDI, and the additional parity/Sudoku examples make the practical value of step-resolved attribution much clearer than in the original submission.

**Key Questions For Authors:**

- In Figure 2B, SDI energy is defined as the sum of absolute stepwise SDI scores over training samples. Although this can be a useful heuristic, have you tried other aggregation or normalization techniques?

- Would it be possible to add runtime and memory table or figure for the scalability section? I think this can strengthen the contributions.

- For the Nanochat experiment, how many checkpoints, seeds, training examples were used to compute the influence summaries? If the conclusions are consistent across seeds or checkpoint choices?

**Limitations:**

yes

**Strengths And Weaknesses:**

## Strengths

The core idea is clean. Once the shared block is unrolled, the body-parameter TracIn score decomposes naturally across recurrent steps, and the conservation identity is a useful check. The sketch-during-backprop implementation get rid of the need for per-example gradient caching is important at transformer scale, making SDI plausibly usable in non-toy settings. Also, Several interesting case studies are introduced.

## Weaknesses

1. The interpretability claims are not fully supported by the evidence. What Figure 1A and Figure 1B show is a correlative pattern between SDI and the observed latent dynamics instead of a causal result. It remains unclear about whether SDI provides a reliable train-data attribution tool.

2. The evaluations seems to miss important baselines. On the attribution side, a systematic comparision against other methods, such as standard scalar TracIn, per-step gradient norms or probe-based analyses could significantly strengthen this paper.

3. The presentation can be improved. As a person who is not familiar with TracIn-style attribution and gradient sketching, I think an illustrative figure of the method could make the paper easier to follow.

---

> ### Author Rebuttal · Authors · 2026-03-28
>
> We would like to thank you for your thoughtful review and address your concerns below.
>
> **Regarding your correlative-vs.-causal concern**, we would like to refer you to our responses to Reviewers pBsP and cKVX. In brief, counterfactual retraining after SDI-guided data removal shows that late-step influential examples are necessary for hard-puzzle performance (removing late-step influential examples degrades hard-puzzle accuracy while affecting easy puzzles less). We also show a specific example of SDI identifying a train sample helping a test sample early in the loop trajectory and becoming harmful later. Thank you for prompting us to include these new experiments.
>
> **Regarding the inclusion of other aggregation or normalisation techniques:** Thank you for this recommendation. We computed the following alternative aggregation methods for the SDI energy curves in Figure 2B in addition to the default sum of stepwise absolute SDI scores across training points, abbreviated as $\sum_i \|\text{SDI}\|$ below: root-mean-square (RMS), top-$k$ mean ($k{=}50$, i.e., averaging the 50 most influential training examples per step), and positive-only sum, which keeps only training examples with positive (helpful) influence, discarding negative (harmful) contributions. These aggregations span the natural summary statistics (L1/L2-norms for $\sum_i \|\text{SDI}\|$ and RMS, robust statistics for top-$k$ and sign-aware for positive-only sum). We also computed a **per-step gradient norm baseline** $\|\varphi_t(z')\|$, the norm of the step-$t$ gradient of the test sample, which measures each step's contribution to the gradient *without* reference to training data.
>
> For each method we compute the Spearman rank correlation $\rho$ of per-example rankings with our default $\sum_i \|\text{SDI}\|$:
>
> |Method|$\rho$ (ours)|
> |-|-|
> |RMS|0.96|
> |Top-50 mean|0.92|
> |Positive sum|0.96|
> |Gradient norm|0.07|
>
> **Findings:**
> - *Robustness to aggregation.* All SDI aggregations produce near-identical per-example rankings ($\rho > 0.92$ with $\sum_i \|\text{SDI}\|$) and all preserve the hard $>$ easy ordering in late-step energy from Figure 2B. The paper's conclusions are not sensitive to the choice of aggregation.
> - *Gradient norms capture different information.* The per-step gradient norm baseline $\|\varphi_t(z')\|$ is uncorrelated with $\sum_i \|\text{SDI}\|$ ($\rho = 0.07$) and *reverses* the hard/easy late-step ordering, confirming that SDI captures training-data-specific information that step-local gradient magnitude alone cannot.
>
> Regarding your requests to compare against standard scalar TracIn: TracIn provides useful aggregate influence (reported in Table 1); the step-resolved analyses in Figure 2 target precisely the regime where SDI extends beyond TracIn's scope, which is our core motivation. Regarding probes, we welcome this suggestion and have conducted a linear probing experiment on Nanochat to determine whether the model indeed learns a "loop counter" as the SDI-based analysis shown in Figure 3 suggests. We find that the loop step is indeed linearly decodable (98.56% accuracy on a held-out split vs. chance accuracy of 25.00%), please also refer to the response to Reviewer uoU9. We believe that this result corroborates the synergy between SDI and probe-based approaches and thank you for the recommendation to include this experiment.
>
> **Regarding your suggestion for a scalability table**, we kindly refer you to our response to reviewer uoU9. In brief, these quantitative results reaffirm that SDI scales favourably even to billion-scale transformers.
>
> **Regarding your request for a figure**: We agree it would help guide reader intuition and will add an overview illustration.
>
> **Regarding NanoChat reproducibility:** The results in the initial manuscript were from a single seed, 82 mid-training checkpoints and 71 SFT checkpoints and we computed SDI on 1024 $\times$ 1024 randomly sampled training/test examples. We now re-ran the experiments with two additional seeds. For *mid-training*, the last-step SDI share is $0.532 \pm 0.010$ at $\tau{=}8$ and $0.506 \pm 0.017$ at $\tau{=}16$. The **last-4-step share** is $0.885 \pm 0.006$ and $0.824 \pm 0.013$ respectively. For *SFT*: last-step share is $0.604 \pm 0.026$ at $\tau{=}8$ and $0.606 \pm 0.028$ at $\tau{=}16$; **last-4-step share** $0.989 \pm 0.002$ and $0.987 \pm 0.001$. These results confirm our initial finding that influence is almost fully allocated to the final loop steps with low between-run fluctuation. The approximately geometric tail is also robust: late-tail log-linear $R^2 \approx 0.996$ for mid-training and $\approx 0.95$ for SFT. We are thus confident in the reproducibility of these results.
>
> We thank you for your careful reading and constructive suggestions, which have meaningfully improved the paper. All new results, the tables and the figure will be added to the camera-ready manuscript. We hope these additions address your concerns and would be happy to discuss further.

---

> > ### Author Rebuttal · Reviewer_rstq · 2026-04-04
> >
> > Thanks for the detailed rebuttal. My main concerns are now adequately addressed. In particular, the new counterfactual retraining results substantially strengthen the causal interpretation of SDI, and the additional parity/Sudoku examples make the practical value of step-resolved attribution much clearer than in the original submission. Overall, these additions materially strengthen both the empirical support and the presentation of the paper. My concerns are fully resolved, and I am raising my score to 4.

---

> > > ### Author Response · Authors · 2026-04-07
> > >
> > > Thank you for your thoughtful suggestions and the constructive discussion!

---

### Official Review · Reviewer_pBsP · 2026-03-12

**Soundness:** 3
**Presentation:** 3
**Significance:** 3
**Originality:** 3
**Overall Recommendation:** 5
**Confidence:** 3

**Summary:**

The paper introduces Step-Decomposed Influence (SDI), a gradient-based data attribution method for weight-tied (looped) transformers. By utilizing a streaming TensorSketch implementation, it unrolls the TracIn estimator across recurrent iterations to localize training data influence to specific computational steps, demonstrating mechanism recovery in algorithmic tasks (Parity, Sudoku) and implicit termination behaviors in a 330M-parameter LLM.

**Compliance With Llm Reviewing Policy:**

Affirmed.

**Final Justification:**

The authors have fully resolved my initial concerns by providing detailed computational breakdowns and additional scaling experiments, demonstrating the method's practicality and solidifying my recommendation.

**Key Questions For Authors:**

1. Does ablating the top decile of "late-step" influential training examples strictly degrade test-time compute scaling on hard Sudoku puzzles without affecting easy ones?

2. Are there concrete empirical instances in the Nanochat or Sudoku setups where $\sum SDI \approx 0$ but $|SDI_t| \gg 0$, and what is the specific mechanistic interpretation of such signal cancellation?

**Limitations:**

yes

**Strengths And Weaknesses:**

**Strengths**

- Streaming TensorSketch Implementation: Resolves the memory bottleneck of per-example gradient materialization. Providing a provably tighter variance bound than prior art enables scalable, step-resolved attribution for large models without prohibitive overhead.

- High-Fidelity State Recovery: The Parity experiment serves as a robust mechanistic sanity check, successfully aligning the SDI gradient trajectories with ground-truth finite-state automaton transitions and validating the metric's descriptive fidelity.

- Empirical Grounding of Test-Time Compute: SDI provides a concrete, observable metric (late-step energy) to explain the performance scaling of looped models on hard tasks (Sudoku), bridging data attribution with dynamic inference algorithms.

**Weaknesses**

- Observational Correlation vs. Causal Mechanism: The framework relies entirely on gradient alignment. It lacks counterfactual data-pruning experiments to prove that ablating "late-step" influential data fundamentally degrades late-step reasoning capabilities, leaving a gap between correlation and causal attribution.

- Linear Decomposition of Non-linear Dynamics: Flattening the highly non-linear BPTT trajectory into an additive sum of step-wise influences risks obscuring dynamic dependencies. It remains unclear whether early-step data structurally alters the latent state initialization to trigger late-step gradient explosions.

- Absence of Signal Cancellation Analysis: Despite motivating SDI as a solution for zero-sum scalar influence (where positive and negative step-influences cancel out), the empirical evaluation focuses exclusively on positive mechanism recovery, omitting analysis of conflicting or toxic sample dynamics.

---

> ### Author Rebuttal · Authors · 2026-03-28
>
> Thank you for your insightful review and your constructive suggestions.
>
> **Regarding Linear Decomposition Risk:** Please note that SDI provably decomposes linearly (Proposition 1) and each summand $\phi_t$ incorporates both the full BPTT signal, which already depends on all later steps and the step-local Jacobian, and thus exactly captures cross-step interactions. Early-step effects that trigger late behavior are therefore still reflected in later $\phi_t$ terms. If desired, Eq. (9) further exposes the full train-step $\times$ test-step SDI matrix with the per-step-decomposed results.
>
> **Regarding observational vs. causal/ablating the top decile of "late-step" examples:** Thank you for this important question! The effect does not yet arise at the top decile level (likely because Sudoku has a lot of inherent structure that is shared between puzzles), but it arises cleanly under a stronger 30% intervention. When we remove the top 30% of training puzzles ranked by late-step SDI mass, we observe a clear counterfactual effect. We computed the ranking on a held-out validation split, checkpoints were selected on that validation split, and final numbers are reported on a disjoint test split of hard puzzles. The key result is that removing the late-step SDI examples somewhat lowers accuracy at late loop steps and/or on easy examples (none are statistically significant), but statistically significantly ($t$-test $p=0.003$) lowers accuracy on hard puzzles. All results are averages +/- STD of 8 runs.
>
> | Condition      | $\tau$=32 overall | $\tau$=64 overall |$\tau$=32, easy (46-47 missing) | $\tau$=32, hard (49-50 missing) |
> |-------------------------------|---------------:|---------------:|------------------:|------------------:|
> | No removal                    | 98.6 +/- 0.7 | 98.9 +/- 0.7 | 98.7 +/- 0.6    | 95.8 +/- 2.7    |
> | Remove 30% late-step SDI examples | 97.8 +/- 0.7 | 98.3 +/- 0.6 | 98.0 +/- 0.8    | **91.7 +/- 1.8**   |
>
> We also tested removing *early-step influential data*, which degrades the model much more broadly, showing high rates of non-convergence (2/8 runs), poor accuracy at all difficulty levels and loop step depths (all <80% accuracy), likely because early-step samples teach the model more "foundational" skills.
>
> We believe this experiment corroborates that SDI provides an added benefit over simpler tools like TracIn. The intervention itself is defined by *when* a training example matters in the recurrence. TracIn can rank examples by total influence, but it cannot formulate a specifically *late-step* removal.
>
> **Regarding your question about signal cancellation:** Signal cancellation, where $\text{TracIn}(z, z') \approx 0$ but $|\text{SDI}_t(z, z')| \gg 0$ at individual steps, is indeed prevalent in both parity and Sudoku and mechanistically interpretable:
>
> - *Parity.* An illustrative example: our dataset contains a 20-bit training sequence and a 40-bit test sequence whose cross-influence is $\text{TracIn} = 0.001$ (suggesting near-irrelevance), yet $\sum_t |\text{SDI}_t| = 17.24$. The SDI trajectory reveals that the 20-bit training example exerts strong *positive* influence during the first $\approx$ 19 loop iterations (cumulative energy +8.62), then *negative* influence for the remaining $\approx$ 20 iterations (cumulative energy −8.62), with the sign flip occurring around step 20, i.e., the length of the training sequence. Mechanistically, the training example teaches useful state transitions for the portion of the recurrence corresponding to its own length, but actively misleads the computation on later loop steps where the test sequence extends beyond positions the training example covers. TracIn collapses this biphasic structure into a near-zero scalar, falsely labelling the example as near-irrelevant.
> - *Sudoku.* To quantify cancellation robustly, define the cancellation ratio $R(z,z') = \sum_t |I_t(z,z')| / |\sum_t I_t(z,z')|$, which equals 1 when all step contributions share the same sign and grows large when positive and negative step influences cancel. Among pairs with substantial step-level influence (top quartile by $\sum_t |I_t|$), the mean cancellation ratio is $\bar{R} = 5.38$ (well above the overall dataset average of 4.3). Moreover, 4.1% of pairs satisfy $R > 5$ (i.e., more than 80% of step-level influence would be lost to cancellation in TracIn) and we observe an interesting pattern: Restricted to pairs where both early-step (1–16) and late-step (17–32) contributions are individually large, the rate of cancellation is 5.0% for the easiest test puzzles and grows to 9.7% for hard test puzzles. We hypothesize that harder puzzles have more complex constraint propagation chains that create opportunities for a training puzzle to help early propagation but hinder late refinement.
>
> We will include all new results in the camera-ready version and thank you again.

---

> > ### Author Rebuttal · Reviewer_pBsP · 2026-04-03
> >
> > I thank the authors for the comprehensive rebuttal. The additional experiments effectively close the logical gaps identified in the initial review.
> >
> > First, the 30% late-step ablation experiment provides the necessary causal evidence that was previously missing (W1/Q1). The statistically significant performance degradation specifically on hard Sudoku puzzles convincingly validates the causal link between late-step influential data and late-step reasoning capabilities.
> >
> > Second, the biphasic influence example provided for the Parity task is a highly effective mechanistic demonstration of signal cancellation (W3/Q2). It empirically proves the limitation of TracIn and justifies the necessity of the SDI formulation. I recommend incorporating this specific case study into the main text of the final version.
> >
> > Finally, the mathematical clarification regarding the retention of cross-step interactions in Eq. (9) resolves my concern regarding linear decomposition (W2).
> >
> > Overall, the rebuttal relies on solid empirical data to address the core critiques. My concerns are fully resolved.

---

> > > ### Author Response · Authors · 2026-04-07
> > >
> > > Thank you for the productive discussion and your insightful review that helped us strengthen our work!

---

### Official Review · Reviewer_uoU9 · 2026-03-13

**Soundness:** 3
**Presentation:** 2
**Significance:** 3
**Originality:** 3
**Overall Recommendation:** 4
**Confidence:** 3

**Summary:**

This paper addresses Looped Transformers, an architecture that performs latent reasoning by repeatedly applying shared parameter blocks. Recognizing that existing training data influence assessment methods (e.g., TracIn) typically provide only an aggregated scalar score and fail to reveal which specific step of the recurrent computation a training sample affects, the authors propose the Step-Decomposed Influence (SDI) framework. SDI unfolds the recurrent computation graph to attribute influence to specific loop iterations. To address the memory bottleneck of computing per-sample gradients on large-scale models, the authors introduce a **TensorSketch**-based implementation that performs sketching directly during backpropagation, avoiding the instantiation of full per-sample gradients.

The experiments demonstrate the effectiveness of SDI across several scenarios: scalability and correctness validation on a 135M-parameter looped GPT; mechanistic interpretability discoveries in parity tasks (revealing finite state automata); analysis of compute scaling laws in Sudoku tasks; and case studies on the Nanochat model.

**Compliance With Llm Reviewing Policy:**

Affirmed.

**Final Justification:**

My concerns have been adequately addressed.

**Key Questions For Authors:**

-	In a single iteration, what percentage of the total time is consumed by TensorSketch projection, FFT operations, and gradient computation, respectively? Where is the primary performance bottleneck?  As the model size scales from 100M to 1B+ parameters, or as the number of loop steps $\tau$ significantly increases, how does the relative time overhead of SDI change? Is there a critical scale beyond which the time cost becomes prohibitive?

-	Given that most modern LLMs are trained with AdamW rather than pure SGD, have you considered incorporating the optimizer's preconditioner matrix into the SDI calculation? If not, could the current dot-product approximation severely underestimate or overestimate the influence of certain samples, particularly for parameters with vastly different adaptive learning rates?

-	In the Nanochat experiments, the observation that influence concentrates in the final few steps suggests the model learned a "step counter." Is there evidence that this mechanism is a byproduct of positional embeddings, or did the model truly learn an internal counting logic? How would this distribution pattern change if positional embeddings were removed?

**Limitations:**

See weakness.

**Strengths And Weaknesses:**

#### **Strengths**
- The paper fills a significant gap in current interpretability literature by providing the first fine-grained attribution tool specifically for internal computation steps in recurrent architectures.
- The proposed "sketch-during-backprop" algorithm using TensorSketch is a significant engineering contribution. Beyond solving the memory bottleneck, the authors derive tighter variance bounds than existing literature, ensuring unbiased and low-variance estimates.
- The experimental results yield fascinating scientific discoveries, such as the identification of limit cycles in parity tasks and the observation that difficult samples retain higher influence in later steps in Sudoku tasks. These findings offer new perspectives on the reasoning mechanisms of looped transformers.
- The method holds strong potential for applications such as calibrating test-time compute, detecting signal cancellation, and targeted data cleaning.

#### **Weaknesses**

-  Lack of Comprehensive Computational Efficiency Analysis (Time-Memory Trade-off):
    *   The paper does not provide an end-to-end training time comparison between the standard training pipeline (or other approximate attribution baselines like sampling-based methods) and the SDI-enhanced workflow. It is unclear how much computational speed is sacrificed to save memory. Does the "space-for-time" trade-off render the method impractical for actual large-scale training?
    *   There is no decomposition of the total overhead into its constituent parts (e.g., backpropagation, TensorSketch projection, FFT computation, aggregation statistics). Without identifying whether the bottleneck lies in FFT or matrix multiplication, future optimization efforts are hindered.
    *   Current experiments focus primarily on a 135M-parameter model. There is a lack of data showing how overhead scales as model parameters increase (from 100M to 1B+) and as the number of loop steps ($\tau$) grows. Does the time overhead grow linearly or quadratically with scale? This is critical for assessing the method's applicability to larger models.

-	SDI relies on TracIn, which is theoretically most interpretable under Stochastic Gradient Descent (SGD). However, modern LLMs often use adaptive optimizers like Adam. It remains debatable whether gradient dot products accurately reflect causal influence under complex optimizer dynamics involving momentum and adaptive learning rates.

-	When training with Truncated Backpropagation Through Time (TBPTT), gradients for early steps may be cut off, resulting in zero SDI for those steps. While the authors mention recomputing with full BPTT, this would exacerbate the aforementioned computational overhead issues, a cost that is not thoroughly discussed.

-	Lack of counterfactual validation: Although SDI can identify influential samples and steps, the paper lacks direct intervention experiments (such as removing high-impact samples or modifying the gradients of specific steps) to causally verify whether these attributions actually change model behavior.

---

> ### Author Rebuttal · Authors · 2026-03-28
>
> We thank you for your constructive review and suggestions and address your concerns below.
>
> **Regarding the Efficiency Breakdown.** Please note that SDI is a post-hoc technique applied to saved checkpoints, so training is not impacted by SDI. The relevant computation time breakdown is SDI gradient featurization vs. a standard forward+backward pass: Gradient computation (fwd+bwd): 51.2%, TensorSketch: 48.4%, SDI inner products: 0.4%. Within each TensorSketch: Scatter-add: 51.8%, FFT: 18.9%, Pointwise multiply + iFFT: 29.2%.
>
> **Regarding Scaling with Model Size.** We did additional scaling experiments with $\tau=32$ on models from 77M to 1.2B parameters (1.2B is FLOP-equivalent to $\approx$ 15B non-looped). All use FP32, $m=2048$, sequence length $L=128$, single A100-80GB. Batch sizes decrease due to memory constraints, leading to lower absolute Sketch time cost. "Fwd+bwd" is a standard forward+backward pass; "Sketch" is the TensorSketch timing. Please note that this time cost is not incurred during training but only once during post-hoc analysis.
>
> |Model|$B$|Fwd+bwd (ms)|Sketch (ms)|Relative sketch cost|
> |-|-|-|-|-|
> |77M|16|454|526|+116%|
> |135M|8|474|447|+94%|
> |386M|4|840|313|+37%|
> |617M|4|1365|354|+26%|
> |1.2B|2|1592|291|+18%|
>
> The relative SDI overhead decreases with model width $d$ because TensorSketch scales as $O(L(d + m\log m))$ while the backward pass scales as $O(Ld^2)$. With fixed $m$, sketch time cost thus only grows linearly in $d$ vs. quadratically for the backward pass.
>
> **Scaling with loop count $\tau$.** 135M model ($B=8$ for $\tau \leq 32$, $B=4$ for $\tau=64$):
>
> |$\tau$|Eff. depth (layers)|Fwd+bwd (ms)|Sketch (ms)|Relative sketch cost|
> |-|-|-|-|-|
> |4|20|84|63|+75%|
> |32|132|474|447|+94%|
> |64|260|564|961|+170%|
>
> Both time costs scale linearly in $\tau$. The relative overhead increases because the Fwd+bwd baseline includes a fixed "non-looped" cost (prelude, coda, embeddings) while TensorSketch is linear in $\tau$. In summary, we do not expect SDI overhead to be the primary bottleneck in post-hoc analysis.
>
> **AdamW Preconditioner.** We compared standard TracIn/SDI with a preconditioned variant using AdamW's second-moment estimate $v$ as suggested: scores become $\eta \cdot (\nabla\ell(z)/\sqrt{v{+}\epsilon}) \cdot (\nabla\ell(z')/\sqrt{v{+}\epsilon})$. We tested the Sudoku model, where the per-coordinate effective learning rate varies by up to $131\times$, and concerns about miscalculated influence are most relevant. Despite this significant spread, *the rankings of influential examples are near-perfectly preserved*: Spearman $\rho = 0.979$ for TracIn, $\rho = 0.934$ for SDI across all steps, and top-5% influential example overlap of 93.7%. Thus, while preconditioning refines absolute magnitudes, it does not significantly alter which examples are identified as influential or when they matter. Note that the preconditioned inner product also preserves the decomposition in Equation (6). We believe that further exploring and optimising preconditioned SDI is a fruitful future research direction.
>
> **On Positional Embeddings vs. Step Counter.** Please note that the model has no explicit loop-step embedding, only standard token positional embeddings. As a preliminary check, we tested whether the step index within the TBPTT window is decodable using a linear probe on activations at a fixed token position (i.e., with the same token positional embeddings). The trained linear probe is able to decode the loop step ($\tau \in [1,4]$ the TBPTT truncation window) 98.56% accuracy on a held-out split while a randomly initialized model with identical architecture achieves exactly chance accuracy of 25.00%. This suggests that the model may indeed possess a kind of “loop counter” that is not trivially explained by that token’s absolute position alone. A similar phenomenon was noted in Geiping et al., 2025, arXiv:2502.05171.
>
> **Regarding the TBPTT concern.** As we note above, SDI is never computed during training, regardless whether TBPTT or full BPTT is used. When the goal is to analyze the optimization actually seen during training, computing SDI with TBPTT at analysis time is the correct choice since the direct step-local SDI term $\phi_t$ outside the truncation window is zero. Even in the setting of full-BPTT recomputation at analysis time (as in our Nanochat experiment), the limiting factor is *not SDI* but computing and storing the activations of the full computational graph.
>
> **Regarding Counterfactual Validation.** We agree that this is an important point and conducted new data ablation experiments for which we refer to our response to Reviewer pBsP. Please also refer to our response to reviewer cKVX regarding individual examples detected through SDI that are “invisible” to standard TracIn.
>
> We will incorporate the new results and discussion in the camera-ready version and are happy to discuss further.

---

> > ### Author Rebuttal · Reviewer_uoU9 · 2026-04-04
> >
> > Thank you for providing the detailed computational breakdown and the additional scaling experiments. The data on operator-level timing effectively addresses my concerns regarding algorithmic efficiency. Furthermore, demonstrating scalability up to a 1.2B parameter model confirms that SDI is a practical post-hoc analysis tool for large-scale models. I will maintain my score.

---

> > > ### Author Response · Authors · 2026-04-07
> > >
> > > Thank you for your thorough review which has meaningfully improved our work!

---

### Official Review · Reviewer_cKVX · 2026-03-13

**Soundness:** 2
**Presentation:** 4
**Significance:** 2
**Originality:** 2
**Overall Recommendation:** 4
**Confidence:** 2

**Summary:**

The authors focus on the concept of using training-data attribution techniques on looped transformers. They investigate the issue of how to break the influence of a training example on a forward pass into an influence per loop iteration, rather than a single aggregate score. The authors test their method on a parity task, a Sudoku task, and a 328M-parameter chat model, examining how influence distributes across loop iterations in each setting. The authors find that per-step trajectories can reveal structure in the latent computation, allowing researchers to better interpret the model's internals.

**Compliance With Llm Reviewing Policy:**

Affirmed.

**Final Justification:**

My main concern was that SDI's utility was minimal and would not reveal insights beyond simpler tools like PCA or scalar TracIn. The authors' rebuttal addressed this by proposing more examples where SDI revealed relevant details about the internal computations of looped transformers. My concerns are fully resolved and I raise my score to 4.

**Key Questions For Authors:**

1. Can you show a specific training example with interpretable influence at a specific loop step? A convincing example would significantly strengthen my assessment of SDI's practical value.
2. Is there a setting where SDI reveals something that simpler tools cannot?
3. Are there any other compelling applications of your method?

**Limitations:**

yes

**Strengths And Weaknesses:**

Strengths
- The paper extends an existing method to work better for a new transformer architecture
- The paper is well-written and has a detailed appendix


Weaknesses:
- Some of the experiments like finding a 4-state cycle in a parity task, does not require data attribution to discover. For example, PCA on hidden states would have revealed the same structure without SDI. It's unclear how counterfactually useful SDI is for generating hypotheses that would yield such findings.
- The mean use case of the method is attributing the computation in particular loops to particular datapoints, but the paper doesn't show any great examples of a specific training example with interpretable influence at a specific loop step.
- The tighter variance bounds are a contribution but have modest practical impact at typical sketch dimensions

---

> ### Author Rebuttal · Authors · 2026-03-28
>
> Thank you for your thoughtful review. We address each of your points below.
>
> **Regarding your concern about the necessity of SDI over simpler tools:** We agree that PCA on hidden states can reveal latent structure. However, while PCA answers *"what does the hidden state look like?"* SDI additionally answers *"which training data is responsible for what happens at each step?"*. We also agree with you that a clearer example of counterfactual hypothesis generation via SDI is helpful, a point that was also raised by Reviewer pBsP. In the response to Reviewer pBsP, we thus provide new evidence from counterfactual retraining experiments that show that SDI identifies training subsets with distinct effects on late-step vs. early-step model performance. This distinction is not directly revealed using TracIn (which flattens the step axis) or PCA (which does not express data influence).
>
> **Regarding your request for specific training examples with interpretable influence at specific loop steps and your question whether SDI can reveal something that simpler tools cannot:** Thank you for the question and recommendation to investigate this. We have conducted additional experiments providing this evidence.
>
> - *Parity.* A characteristic example from our dataset is a 20-bit training sequence and a 40-bit test sequence where TracIn cross-influence is near-zero $(0.001)$ suggesting irrelevance, but the SDI absolute sum is very large $\left(\sum_t |\text{SDI}_t| = 17.24\right)$. The SDI trajectory reveals that the training example exerts strong *positive* influence during the first $\approx 19$ loop iterations, then *negative* influence for the remaining $\approx 20$ iterations, with the sign flip occurring around step 20, i.e., the length of the training sequence. Mechanistically, the training example teaches useful state transitions for the portion of the recurrence corresponding to its own length, but misleads the computation on later steps where the test sequence extends beyond positions the training example covers. TracIn reports a near-zero aggregate, masking this biphasic structure and falsely suggesting that the training sample is irrelevant.
>
> - *Sudoku.* For hard test puzzles ($\geq 48$ missing), the most influential training puzzle at early steps (1–4) is a *different* training example from the most influential at late steps (29–32) in **90.4%** of cases. For instance, for a specific test puzzle, the early-dominant training puzzle might contribute strongly during the initial solving stage while the late-dominant training puzzle exerts its influence specifically during the late "iterative refinement" stage. TracIn tells you *which* training puzzle matters for the final task accuracy; only SDI tells you *when* and reveals that different training examples drive different stages of the model's computation.
>
> **Regarding your question about other compelling applications:**
> The parity example above demonstrates a general-purpose debugging application: SDI identifies training examples whose aggregate TracIn is near-zero but whose step-resolved trajectory reveals large opposing effects. In production, such examples may silently corrupt specific computational phases while appearing benign under scalar attribution. Moreover, the SDI energy curves (Figure 2B) and results in the response to Reviewer pBsP characterize *when* additional loop iterations cease to contribute meaningful influence for a given difficulty class. This could inform an instance-wise early-stopping criterion, i.e., halting the recurrence when SDI energy drops below a threshold without needing an expensive grid search over the loop count.
> More speculatively, a promising use-case is using SDI as a *diagnostic for RLHF and alignment:* As looped architectures are adopted in RLHF and preference-optimization settings, SDI can decompose the influence of preference pairs across recurrent steps, revealing whether alignment data primarily shapes early instruction-following dynamics or late reasoning/refinement steps. This could help diagnose subtle alignment failures that emerge specifically in late-step computation, a regime where scalar attribution would aggregate away the signal. We regard this as a natural and important extension and intend to investigate it in future work.
>
> **Regarding your concern about the variance bounds:** We would like to note that our novel variance bounds apply not just to SDI, but to the TensorSketch algorithm itself. TensorSketch is an important building block of kernel approximation algorithms, streaming algorithms and many numerical linear algebra techniques. We believe our theoretical results to therefore be of independent interest to the community. Moreover, our bounds are provably tight (Lemma 4), establishing the fundamental limit for TensorSketch estimation.
>
> We will incorporate these results into the camera-ready manuscript and would be happy to discuss further.

---

> > ### Author Rebuttal · Reviewer_cKVX · 2026-04-04
> >
> > My concerns are resolved. I raised my score to 4.

---

> > > ### Author Response · Authors · 2026-04-07
> > >
> > > Thank you for your constructive feedback and the productive discussion!

---

### Decision · Program_Chairs · 2026-04-30

**Decision:**

Accept (regular)

**Comment:**

This paper introduces Step-Decomposed Influence (SDI), a method for decomposing training-data influence across recurrent steps in looped transformers, together with a scalable TensorSketch implementation. Reviewers agreed that the paper is technically sound, clearly written, and makes a novel and useful contribution to training-data attribution for looped architectures.

The main concerns in the initial reviews were about significance and validation, rather than soundness. The reviewers questioned whether SDI provides insights beyond simpler tools (e.g., PCA or TracIn), whether the results were mainly correlative or whether computational costs were well characterised. After reading the rebuttal and discussion, I believe the evidentiary picture changed. The authors provided clear signal-cancellation examples where aggregate TracIn is near zero but SDI reveals strong step-wise effects, and added counterfactual data-removal experiments showing targeted performance degradation. Together, I believe, these results address the concern that simpler methods could explain the findings and strengthen the causal interpretation. The rebuttal also clarified computational tradeoffs.

A great discussion between authors and reviewers; the additions by the authors resolved the main concerns raised by the reviewers, all of whom indicated after the rebuttal that their concerns were addressed. Some weaknesses remain e.g., strongest results appearing in the rebuttal and limited comparisons, but they do not outweigh the strengths. As such, I recommend acceptance. The paper is technically sound, non-redundant, and useful to the ICML community.